# High-throughput fluorescence lifetime imaging flow cytometry

Hiroshi Kanno [1,2] ✉, Kotaro Hiramatsu [1,3], Hideharu Mikami [1,4], Atsushi Nakayashiki[5], Shota Yamashita[5], Arata Nagai[5], Kohki Okabe[6], Fan Li[1], Fei Yin[2], Keita Tominaga[5], Omer Faruk Bicer[1], Ryohei Noma[7], Bahareh Kiani[8], Olga Efa[8], Martin Büscher [8], Tetsuichi Wazawa[7], Masahiro Sonoshita[9], Hirofumi Shintaku[10], Takeharu Nagai [7], Sigurd Braun [11], Jessica P. Houston [12], Sherif Rashad[2,13], Kuniyasu Niizuma [2,5,13] & Keisuke Goda [1,14,15] ✉

Flow cytometry is a vital tool in biomedical research and laboratory medicine. However, its accuracy is often compromised by undesired fluctuations in fluorescence intensity. While fluorescence lifetime imaging microscopy (FLIM) bypasses this challenge as fluorescence lifetime remains unaffected by such fluctuations, the full integration of FLIM into flow cytometry has yet to be demonstrated due to speed limitations. Here we overcome the speed limitations in FLIM, thereby enabling high-throughput FLIM flow cytometry at a high rate of over 10,000 cells per second. This is made possible by using dual intensity-modulated continuous-wave beam arrays with complementary modulation frequency pairs for fluorophore excitation and acquiring fluorescence lifetime images of rapidly flowing cells. Moreover, our FLIM system distinguishes subpopulations in male rat glioma and captures dynamic changes in the cell nucleus induced by an anti-cancer drug. FLIM flow cytometry significantly enhances cellular analysis capabilities, providing detailed insights into cellular functions, interactions, and environments.

Flow cytometry[1–3] serves as a cornerstone of biomedical research and laboratory medicine, enabling high-throughput analysis of the physical and chemical characteristics of single cells. It has been used for various purposes, including blood testing[4], rare cell detection[5], cell cycle analysis[6], and drug screening[7] in settings ranging from basic research to clinical applications. Flow cytometry primarily measures the fluorescence intensity of fluorophores to quantify cell surface antigens or proteins for cell type identification[4,5] and to assess changes in physiological parameters such as pH and calcium ion concentration[8]. However, its accuracy can be compromised by fluctuations in fluorescence intensity due to factors such as light scattering and absorption within complex cellular structures, concentration dependence and overlapping emission spectra of fluorescent molecules, and inconsistent excitation light intensity[9]. In particular, intercellular and

[1]Department of Chemistry, The University of Tokyo, Tokyo, Japan. [2]Department of Neurosurgical Engineering and Translational Neuroscience, Tohoku University Graduate School of Medicine, Miyagi, Japan. [3]Department of Chemistry, Kyushu University, Fukuoka, Japan. [4]Research Institute for Electronic Science, Hokkaido University, Hokkaido, Japan. [5]Department of Neurosurgery, Tohoku University Graduate School of Medicine, Miyagi, Japan. [6]Graduate School of Pharmaceutical Sciences, The University of Tokyo, Tokyo, Japan. [7]SANKEN (The Institute of Scientific and Industrial Research), Osaka University, Osaka, Japan. [8]Miltenyi Biotec B.V. & Co. KG, Bergisch Gladbach, Germany. [9]Institute for Genetic Medicine, Hokkaido University, Hokkaido, Japan. [10]Institute for Life and Medical Sciences, Kyoto University, Kyoto, Japan. [11]Institute for Genetics, Justus-Liebig-University Giessen, Giessen, Germany. [12]Department of Chemical and Materials Engineering, New Mexico State University, Las Cruces, NM, USA. [13]Department of Neurosurgical Engineering and Translational Neuroscience Graduate School of Biomedical Engineering, Tohoku University, Miyagi, Japan. [14]Institute of Technological Sciences, Wuhan University, Hubei, China. [15]Department of Bioengineering, University of California, Los Angeles, CA, USA. ✉e-mail: hkanno@g.ecc.u-tokyo.ac.jp; goda@chem.s.u-tokyo.ac.jp

intracellular variations in fluorophore concentrations are unavoidable when dealing with highly heterogeneous cells (e.g., tumor cells)[10], which critically hinders accurate interpretations of flow-cytometric data, even with the assistance of imaging (i.e., fluorescence imaging flow cytometry[11–13]).

Fluorescence lifetime measurement effectively bypasses these precision issues, as fluorescence lifetime remains largely unaffected by light scattering, absorption, excitation light intensity variability, and fluorescent molecule concentration[14]. It further benefits from its ability to distinguish different fluorophores and exhibits sensitivity to environmental factors including action potential[15], temperature[16], ion concentration[17], and molecular interactions[18,19]. While fluorescence lifetime measurement has been integrated into flow cytometry[20], the absence of spatial information in fluorescence lifetime flow cytometry hampers its ability to fully examine environmental influences on multidimensional cellular structures, including localized molecules and the morphology of organelles. To overcome this limitation, attempts have been made to integrate high-speed fluorescence lifetime imaging microscopy (FLIM) into flow cytometry[21] or flow particle analysis[22]. However, these efforts faced significant obstacles due to the requirement for picosecond light pulses[21,22], >10 GHz bandwidth modulation of a swept-source Fourier-domain Mode-locked laser[21], and high-speed photodetectors with sub-nanosecond temporal resolution[21] or even shorter (i.e., streak camera)[22] to resolve single fluorescence decays. Additionally, even with these complex components involved, the previous demonstrations were preliminary and limited to only capturing fluorescence lifetime images of a relatively small number of objects (<100) flowing at low speeds (≤0.2 m/s). This level of throughput falls short of the requirements for statistically significant cell analysis, which in biomedical research and laboratory medicine typically requires examining over 10,000 cells[1–3]. On the other hand, FLIM can be conducted in the frequency domain using a continuous-wave laser[17,23,24], but its imaging speed is limited to at most the video rate[17], rendering practical FLIM flow cytometry unfeasible. To the best of our knowledge, there has been no demonstration of a continuous, high-throughput FLIM image acquisition system that can match traditional flow cytometry in terms of large-scale image analysis, practical applicability, or the potential biomedical insights it could offer.

In this research article, we introduce a significant advancement in FLIM speed, surpassing previous limits of 1000–2000 frames per second[15,21], thereby enabling high-throughput FLIM flow cytometry. Our FLIM flow cytometer employs a continuous-wave laser as a light source, modulates it within a bandwidth of only <222 MHz, generates dual intensity-modulated beam arrays with complementary modulation frequency pairs, and uses them to enable the continuous and precise fluorescence lifetime image acquisition of rapidly flowing cells. Remarkably, we achieve an event rate exceeding 10,000 events per second (eps) at high flow speeds of at least 3 m/s (where an event is defined as a single cell, a cell cluster, or cell debris)[1,2]. Our system demonstrates strong resilience to fluctuations in fluorescence intensity and high sensitivity to environmental factors. This is evidenced by analyzing extensive fluorescence lifetime images of various samples, both non-biological and biological, with high statistical significance. Moreover, our FLIM flow cytometry proves its immediate value in biomedical research; it distinguishes subpopulations in male rat glioma and captures dynamic changes in the cell nucleus induced by an anti-cancer drug. FLIM flow cytometry marks a significant advancement in cellular analysis, offering in-depth understanding of cellular functions, interactions, and environments.

## Results

### Principles of high-throughput FLIM flow cytometry

Our FLIM design effectively multiplexes the acquisition of fluorescence lifetime pixels in frequency-domain FLIM[17,23,24] by employing dual intensity-modulated continuous-wave beam arrays, enabling the simultaneous acquisition of bright-field, fluorescence intensity, and fluorescence lifetime images of rapidly flowing cells on a microfluidic chip (Fig. 1a–c). These dual intensity-modulated beam arrays, each equipped with stepwise modulation frequencies ranging from 21 to 222 MHz at intervals of 4.1 MHz, serve as the excitation light for the flowing cells[25–30] (Supplementary Fig. 1). Each beam, meticulously focused by an objective lens, produces an intensity-modulated fluorescence signal exhibiting a phase delay relative to the excitation (Fig. 1b), which is used to obtain fluorescence lifetime information. This phase delay at a specific time point is expressed as

$$\theta_{k,j} = \arg\left(-\sum_l \frac{\alpha_{k,j}^l}{1 + i\omega_k \tau_{k,j}^l}\right), \tag{1}$$

where $\omega_k$, $\tau_{k,j}^l$, and $\alpha_{k,j}^l$ represent the angular frequency of the $k$th beam spot, the $l$th mono-exponential fluorescence lifetime component at the spatial point being scanned at the $j$th time point, and its intensity-weighted fractional contribution, respectively, with $\sum_l \alpha_{k,j}^l = 1$ and $i^2 = -1$. The generated fluorescence arrays, after being spatially separated at the intermediate image plane by a pickoff mirror, are detected by distinct avalanche photodetectors (Fig. 1a). The signals, although multiplexed from the fluorescence array via single-pixel photodetection, undergo demultiplexing and demodulating through post-detection digital signal processing due to their unique modulation frequency bands[25,26] (see Methods section and Supplementary Fig. 2).

A distinctive feature of our approach is the spatial inversion implemented on one of the beam arrays. This scheme enables the interrogation of a flowing cell's identical spatial point with a complementary modulation frequency pair ($f_{low}$ and $f_{high}$), generating amplitude and phase image pairs that are inverted concerning the center frequency of $(f_1 + f_n)/2$ (Fig. 1c). This pair of phase images ensures precise fluorescence lifetime measurements, particularly for cases demonstrating non-mono-exponential decay[23] (detailed in the Methods and Discussion section). Similarly, the pair of amplitude images enhances the sensitivity of fluorescence intensity imaging. Consequently, this FLIM scheme, using only a 222 MHz bandwidth, allows for the concurrent acquisition of both fluorescence intensity and lifetime images (up to 50 × 192 pixels) of cells moving at speeds between 1–3.5 m/s with a high spatial resolution of up to 0.8 μm (Fig. 1d–g).

### Demonstration of FLIM flow cytometry at over 10,000 eps

To experimentally demonstrate the high-throughput capability of the FLIM flow cytometer, we measured 6-μm beads (Polysciences 17156-2) and live cells (Jurkat) stained with Calcein-AM flowing on the microfluidic chip at high flow speeds (over 3 m/s). Specifically, we repeated the acquisition of 10,000 consecutive events (each data acquisition time: 0.82–0.89 seconds) triggered by detecting fluorescence signals emitted from the flowing objects (Fig. 2a and Supplementary Fig. 3) and calculated the event rate (detailed in the Methods section). Consequently, we found that the actual event rates for the beads and cells were 11,297 eps and 10,371 eps, respectively[31] (Fig. 2b). These event rates were at least one order of magnitude higher than the highest value reported to date[21] (although the claimed event rates were physically unattainable based on the reported flow speed and conventional event rate definition). Additionally, we confirmed that the triggered events followed the Poisson distribution for both beads and cells (Fig. 2c)[25,32]. The absence of time intervals from 0 to 20 μs was due to the data length of each event (20.48 μs). Moreover, the abundance of time intervals around 20.5 μs suggests that a larger number of events were triggered once the FLIM flow cytometer was ready for the next data acquisition after completing the previous one. This was primarily attributed to the high sample concentrations (~$10^7$ objects/mL), suggesting that the

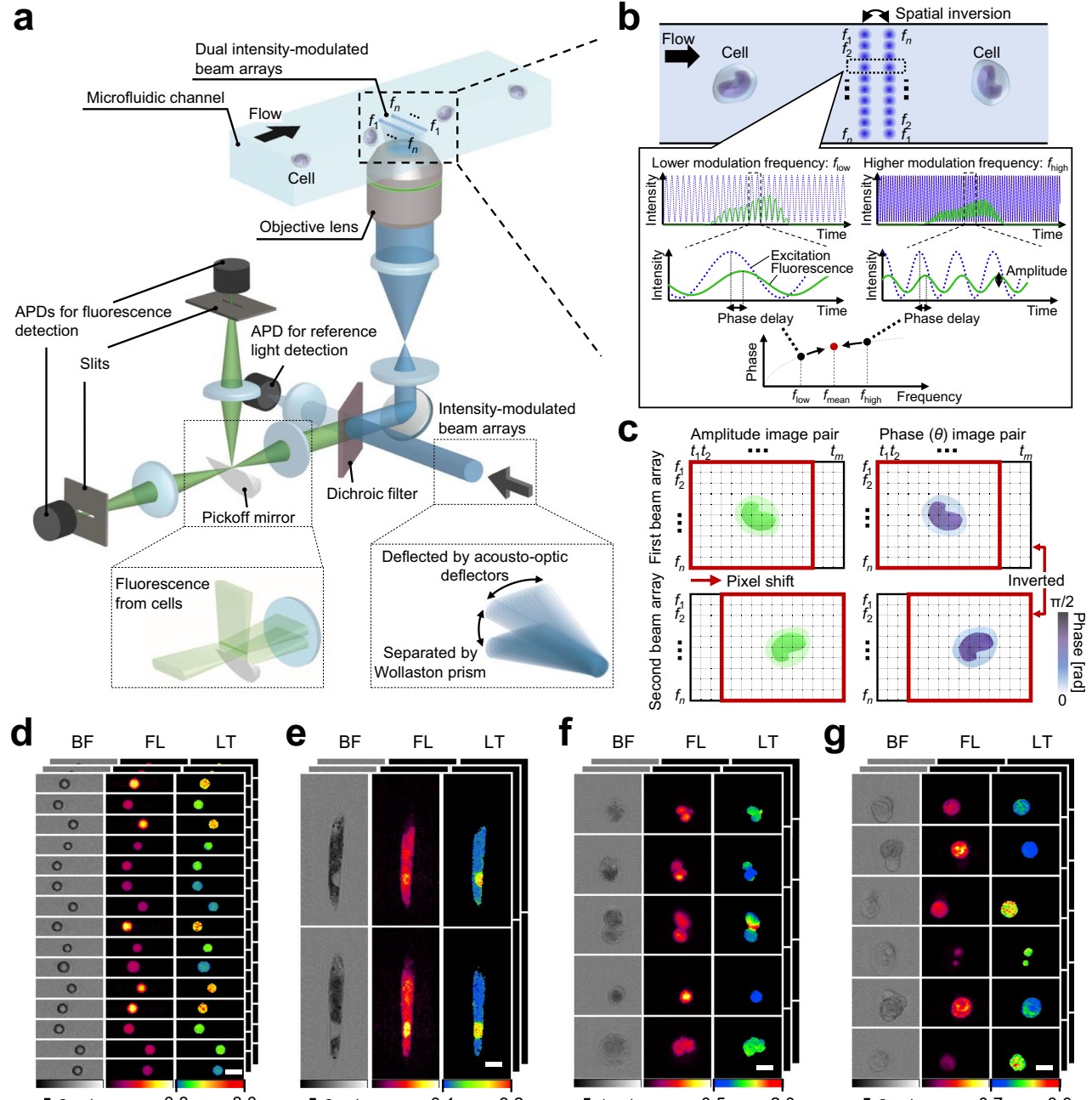

**Fig. 1 | High-throughput FLIM flow cytometry.** $f_1$–$f_n$ denote modulation frequencies ($n$: the total number of modulation frequencies). **a** Schematic of the high-throughput FLIM flow cytometer. Dual intensity-modulated beam arrays enable double interrogation of a flowing cell, enhancing the precision of FLIM image acquisitions. APD: avalanche photodetector. **b** Principle of fluorescence lifetime pixel acquisition with a complementary modulation frequency pair ($f_{low}$ and $f_{high}$). The time-varying phase delay of the fluorescence signal relative to the excitation was used for generating fluorescence lifetime images. $f_{mean} = (f_{low} + f_{high})/2 = (f_1 + f_n)/2$.

**c**, Amplitude and phase image pairs generated from intensity-modulated beam arrays. $t_1$–$t_m$ represent time points ($m$: the number of time points acquired for each triggered event). **d**–**g** Images of beads and cells acquired with the FLIM flow cytometer. BF bright-field, FL fluorescence, LT fluorescence lifetime. Scale bars: 10 μm. Two, two, four, and five independent experiments were performed, resulting in similar results for panels **d**–**g**, respectively. **d** Polymer beads with different fluorescence lifetimes. **e** *Euglena gracilis* cells stained with SYTO16. **f** Tumor-derived rat glioma cells stained with SYTO16. **g** Human cancer cells (Jurkat) stained with SYTO16.

demonstrated event rates approached the upper limits attainable with the FLIM flow cytometer.

Next, we statistically evaluated the obtained 10,000 images, from which the beads ($n = 9659$) and the cells ($n = 9036$) were extracted for analysis. We found that fluorescence lifetimes for beads and cells were $1.66 \pm 0.06$ ns [mean ± standard deviation (SD)] and $2.78 \pm 0.26$ ns, respectively (Fig. 2d), almost consistent with the literature value of Calcein[33]. The negative correlation between fluorescence lifetime and fluorescence intensity for the cells can be attributed to the self-

quenching of fluorescent molecules[34]. Additionally, comparing the coefficient of variations (CVs) for fluorescence intensity and fluorescence lifetime in Fig. 2d, the fluorescence lifetime showed smaller variation between objects than the fluorescence intensity for both samples. Similarly, comparing the CVs of the fluorescence intensity and fluorescence lifetime pixel values within each object, the fluorescence lifetime exhibited smaller variation for both samples (Fig. 2e). These results indicate that FLIM flow cytometry is robust against factors affecting not only inter-object but also intra-object fluorescence

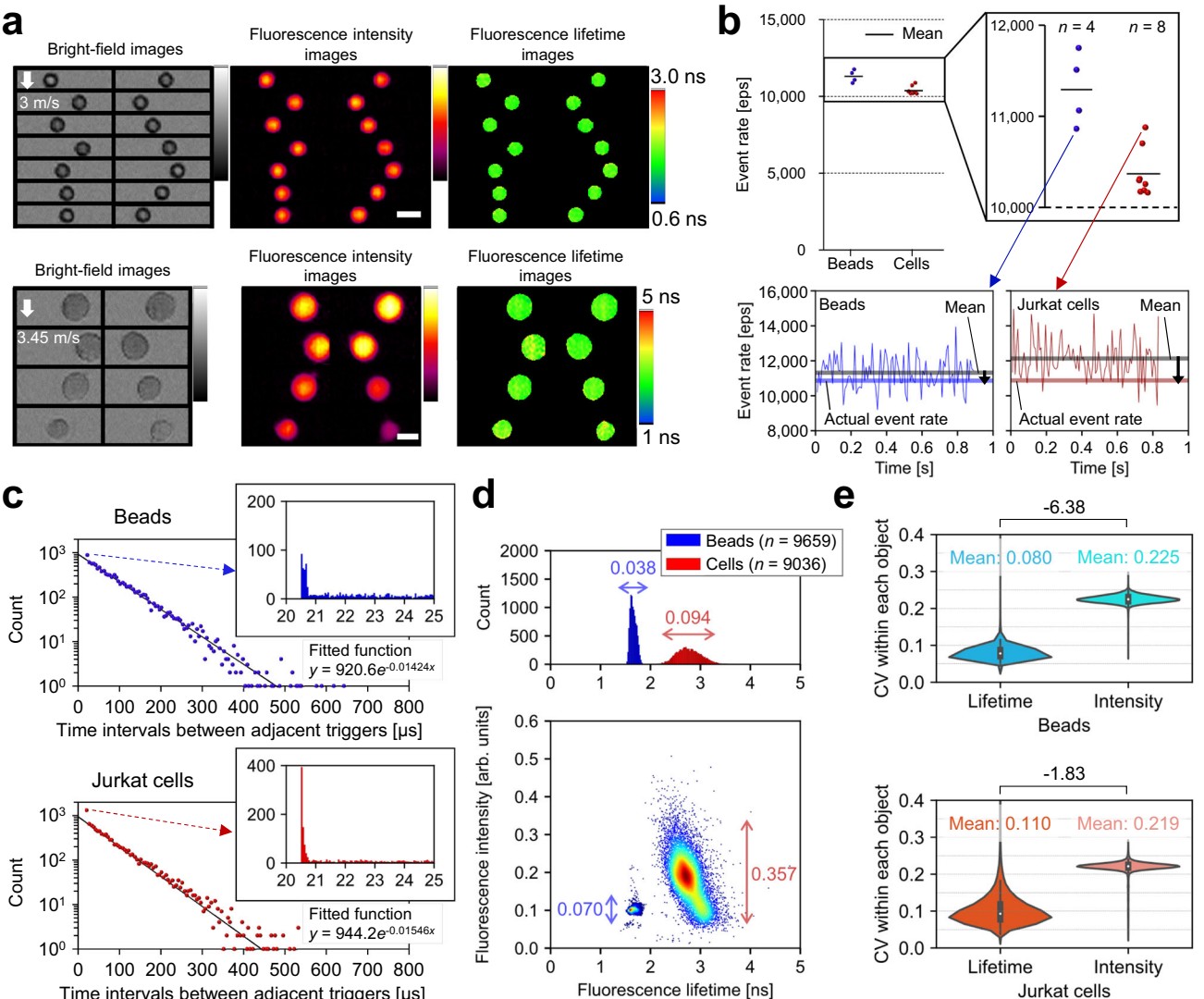

**Fig. 2 | Demonstration of FLIM flow cytometry at over 10,000 eps.** The panels (**a**, line graphs in **b**, **c**–**e**) were generated from one of the image acquisition trials of 10,000 consecutively triggered events, from which 6-µm fluorescent beads (n = 9659) and Jurkat cells (n = 9036) were extracted (Supplementary Fig. 3). **a**, Representative images of 6-µm beads obtained at 10,862 eps and of Jurkat cells stained with Calcein-AM obtained at 10,877 eps. Scale bars: 10 µm. **b** Demonstrated event rates. The line graphs show the relationship between the instantaneous (per 100 events), mean, and actual event rates calculated from 10,000 event acquisitions. n represents the number of image acquisition trials conducted. **c** Distributions of time intervals (n = 9999) between adjacent trigger events for beads (top, blue) and cells (bottom, red). Each dot in the scatter plots represents the total number of time intervals within every 5 µs as the vertical value and their

average as the horizontal value. Histograms show the details of the dots representing 20–25 µs time intervals. Solid lines are exponential fits to the dots based on the least absolute residuals. **d** Distributions of beads and cells in fluorescence intensity and fluorescence lifetime. The double-headed arrows indicate the coefficient of variations (CVs) for fluorescence intensity and fluorescence lifetime. **e** Distributions of beads (n = 9659) and cells (n = 9036) in the CVs of fluorescence intensity and fluorescence lifetime pixels within each object. Difference values between two samples represent effect sizes (Cohen's d), with their standard errors less than 0.04. The violin plots display the median values with white dots, the first and third quartiles with box edges, and 1.5 times the interquartile range (IQR) with whiskers. Source data are provided as a Source Data file.

intensity variations, compared to fluorescence intensity imaging flow cytometry.

## Differentiation of inter-object fluorescence lifetime components

To characterize the capability of the FLIM flow cytometer for distinguishing inter-object fluorescence lifetime components, we measured a mixture of 6.5-µm polymer beads, each characterized by fluorescence lifetime values of 1.72 ns, 2.71 ns, and 5.54 ns (PolyAn 11000006, 11001006, 11002006)[35]. The images, pseudo-colored in fluorescence lifetime values, revealed the presence of the three types of beads (Fig. 3a), which was supported by the clear subpopulations in the scatter plot (Fig. 3b). Additionally, these beads exhibited three

well-separated peaks in the fluorescence lifetime histogram while two out of the three peaks overlapped in fluorescence intensity. This emphasizes the limitation of distinguishing these beads solely based on fluorescence intensity measurements including traditional flow cytometry. Moreover, differentiation of these polymer beads based on their fluorescence wavelengths was not feasible due to the spectral overlap in the 550–800 nm range[35]. Thus, fluorescence lifetime was the sole effective parameter for this classification. Notably, the fluorescence-lifetime-based classification was not impacted by doublet beads or inhomogeneous fluorescent staining, as indicated by the arrows in Fig. 3b (see Supplementary Fig. 4 for the scatter plot obtained by measuring each type of beads separately). After segregating the three subpopulations of the beads using gates at 1.6 ns and

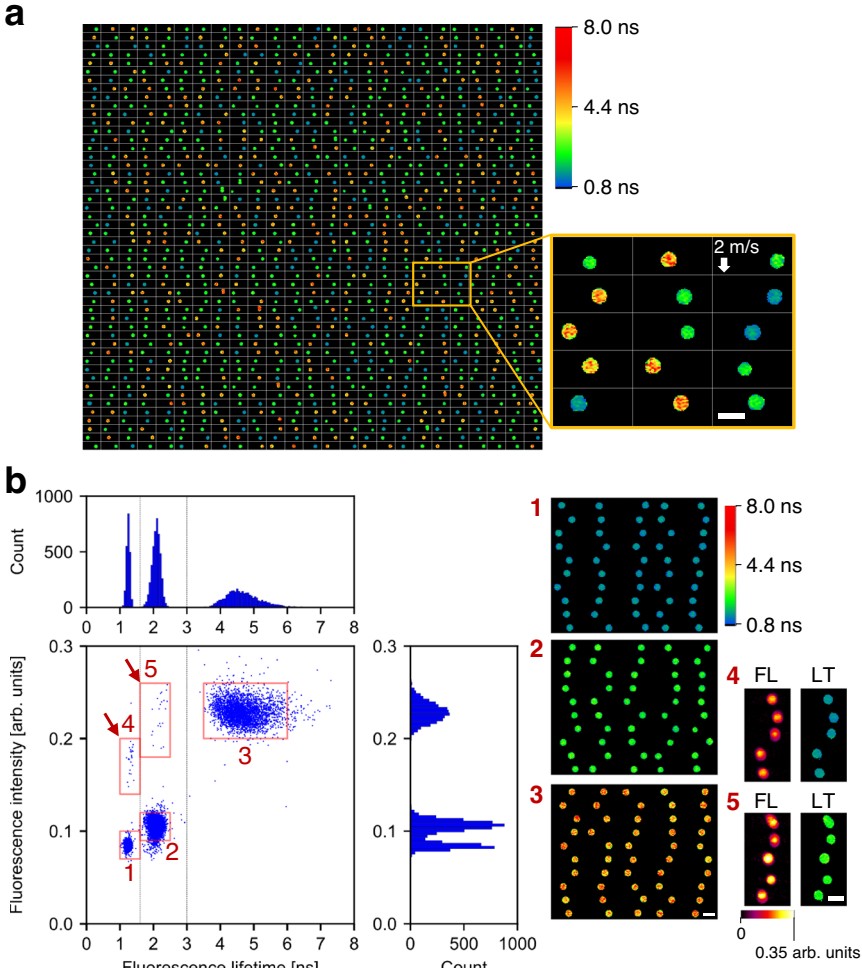

**Fig. 3 | Differentiation of inter-object fluorescence lifetime components by FLIM flow cytometry.** Scale bars: 10 μm. **a** Representative fluorescence lifetime images (50 rows × 25 columns) obtained from a mixture of 6.5-μm polymer beads with fluorescence lifetime values of 1.7 ns, 2.7 ns, and 5.5 ns. The flow speed was 2 m/s. Two independent experiments were performed, resulting in similar results. **b** Distributions of the fluorescent beads (*n* = 9945). Dashed lines represent 1.6 ns and 3.0 ns. The right images show the beads randomly picked from the corresponding red boxes in the scatter plot. FL fluorescence intensity, LT fluorescence lifetime. Source data are provided as a Source Data file.

3.0 ns, we calculated the mean and SD of the fluorescence lifetimes for each section. The resulting values were 1.25 ± 0.05 ns, 2.09 ± 0.13 ns, and 4.70 ± 0.54 ns (mean ± SD), which were closely aligned with, albeit slightly smaller than, the nominal values (see Discussion section for the discrepancy).

### Differentiation of intra-object fluorescence lifetime components

To characterize the ability of the FLIM flow cytometer to distinguish the intracellular environmental differences, we analyzed the fluorescence lifetime of *Euglena gracilis* cells (a single-celled microalgal species) stained with SYTO16 for labeling their nuclei. While the images revealed the presence of multiple conditions of *E. gracilis* cells (Fig. 4a), we specifically selected the elongated cells with an eccentricity exceeding 0.95 for further analysis. Their fluorescence lifetime images (Fig. 4b) and the corresponding scatter plots (Fig. 4c) clearly illustrate that SYTO16 in the nuclei shows longer fluorescence lifetimes than autofluorescence from intracellular chlorophyll, irrespective of fluorescence intensity values. Additionally, the fluorescence lifetime of chlorophyll was estimated from the histogram of fluorescence lifetime image pixels (Fig. 4d), resulting in a value of 0.34 ± 0.17 ns (mean ± SD), aligning with a literature value[36]. Moreover, to statistically distinguish intracellular local environments based on fluorescence lifetime

measurements, we partitioned the acquired fluorescence lifetime images (*n* = 1290) with a cutoff value of 0.9 ns and extracted, via CellProfiler[37], morphological features of areas with fluorescence lifetime >0.9 ns (considered equivalent to the nucleus region) and areas with fluorescence lifetime <0.9 ns (considered equivalent to the chloroplast region) in the *E. gracilis* cells. Consequently, both areas exhibited distinctive values in several morphological parameters that characterize the intracellular structure and composition of the cells (Fig. 4e). This underscores the power and effectiveness of FLIM flow cytometry in assessing intracellular local environments of many cells.

### Investigation of cellular heterogeneity in rat glioma

Investigating the phenotypic diversity of cells within a malignant tumor in vivo is crucial for understanding the pathological features of the tumor[38]. In this application, we used high-throughput FLIM flow cytometry to characterize intratumor phenotypic heterogeneity in rat glioma. Specifically, we induced tumor formation by implanting rat glioma cells (GS-9L) into a male rat brain, allowed them to develop for 18 days, dissociated the removed tumor, and stained the nuclei, followed by subsequent analysis with FLIM flow cytometry (detailed in the Methods section) (Fig. 5a and Supplementary Fig. 5). As shown in Fig. 5b, while flow cytometry (black

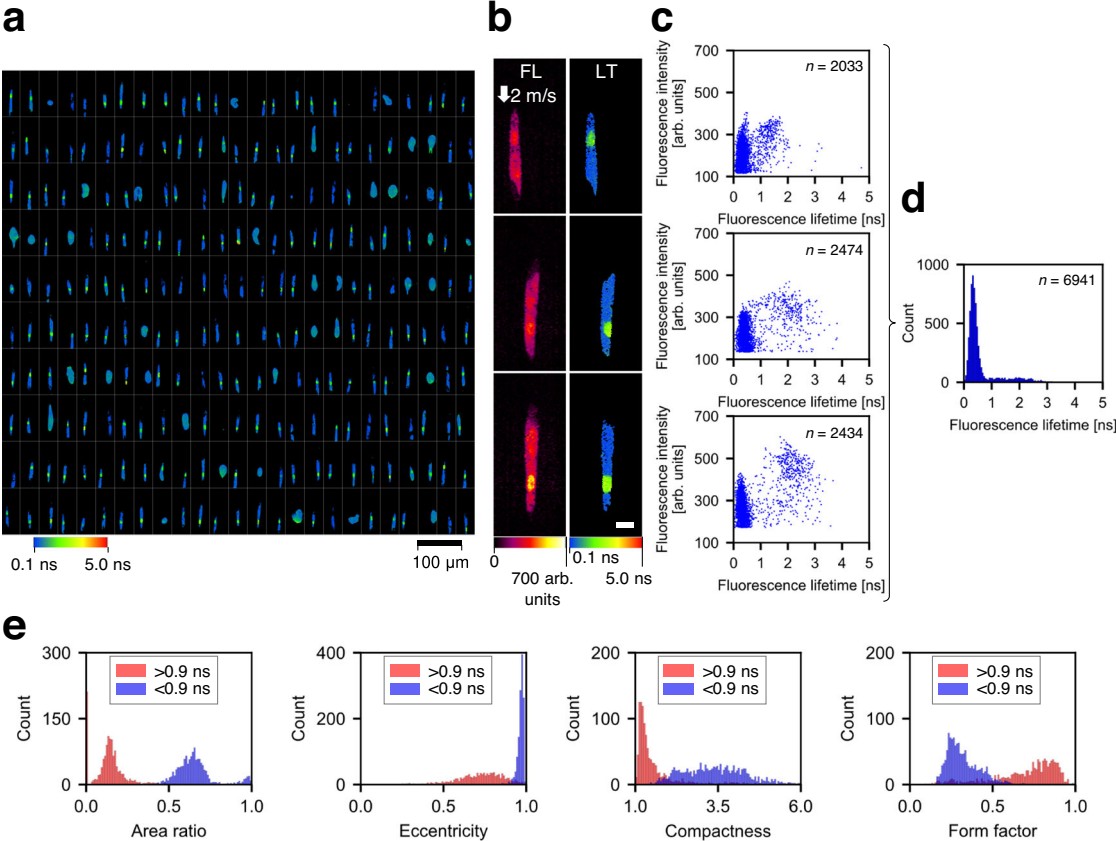

**Fig. 4 | Differentiation of intra-object fluorescence lifetime components by FLIM flow cytometry. a** Representative fluorescence lifetime images (10 rows × 25 columns) of *E. gracilis* cells stained with SYTO16 and flowing at 2 m/s. The images were segmented by a specific threshold value of fluorescence intensity (160 arb. units). Only *E. gracilis* cells with elongated shapes (eccentricity > 0.95) were used for subsequent analyses (**b**–**e**). **b** Representative images of *E. gracilis* cells with an eccentricity of >0.95, which includes images in Fig. 1e. FL fluorescence intensity, LT fluorescence lifetime. Scale bar: 10 μm. Two independent experiments were performed, resulting in similar results for panels **a** and **b**.

**c** Scatter plots showing pixel distributions in fluorescence intensity and fluorescence lifetime corresponding to the images in panel **b**. *n* represents the number of image pixels. **d** Histogram of the pixel values from the fluorescence lifetime images in panel **b**. *n* represents the number of image pixels. **e** Representative morphological features of objects occupied with <0.9-ns pixels (blue, *n* = 1290) and with >0.9-ns pixels (red, *n* = 1079). The area ratio is defined as the ratio of an object area with <0.9 ns or > 0.9 ns to the entire cell area. Source data are provided as a Source Data file.

histogram) and fluorescence lifetime flow cytometry (blue histogram) observed only one and two subpopulations, respectively, FLIM flow cytometry revealed a third subpopulation which derived from intranuclear variations indicated by the fluorescence lifetime gradient from nuclear periphery to center (Supplementary Fig. 6). Given that the cultured glioma cells did not show these subpopulations (Supplementary Fig. 7), they could result from interactions with diverse tumor microenvironments. To delve deeper, we analyzed the morphological features of cells and their nuclei for each subpopulation (Fig. 5c). Consequently, the subpopulations exhibited variations in morphological features such as size, compactness, and eccentricity. Specifically, the first subpopulation (SP1) showed overall smaller variations. In contrast, the second (SP2) and the third subpopulations (SP3) showed increased morphological complexity, as indicated by factors including the percentage of cells with multiple nuclei (see Methods to refer to calculation methods used). Additionally, the SP3 exhibited larger cell and nucleus areas. These results indicate that FLIM flow cytometry directly probed the diversity of intranuclear physical states within a single tumor, originating from various cell types including glioma cells and immune cells (e.g., lymphocytes and macrophages)[39]. Remarkably, this separation of the subpopulations, achieved without the use of antibodies (e.g., CD markers), remained unattainable using any combination of fluorescence intensity, fluorescence lifetime, or morphological

features of cells and their nuclei (Supplementary Fig. 8), suggesting the capability of FLIM flow cytometry in detecting cellular characteristics unidentified with conventional flow cytometry and imaging flow cytometry.

## Large-scale analysis of temporal nucleus dynamics induced by an anti-cancer drug

Precise assessment of drug efficacy is essential for optimizing cancer treatment outcomes[40,41]. In this application, we used FLIM flow cytometry to monitor temporal changes in the nucleic state of human cancer cells (Jurkat) induced by an anti-cancer drug. Specifically, we conducted FLIM flow cytometry of Jurkat cells after treating them with doxorubicin for durations ranging from 0 min (i.e., control) to 140 min and staining their nuclei with SYTO16. (detailed in the Methods section) (Fig. 6a, Supplementary Figs. 9, and 10). While conventional flow cytometry failed to detect significant changes, as indicated by small effect sizes (Cohen's $d < 0.25$) in both cell and nucleus areas, fluorescence lifetime flow cytometry and FLIM flow cytometry revealed substantial transitions (Cohen's $d > 0.5$) in fluorescence lifetime and fluorescence lifetime gradient, respectively (Fig. 6b). Notably, these transitions occurred at different time intervals (i.e., 0–50 min for fluorescence lifetime flow cytometry and 50–110 min for FLIM flow cytometry). This suggests that doxorubicin initially permeated the entire nuclei and subsequently induced

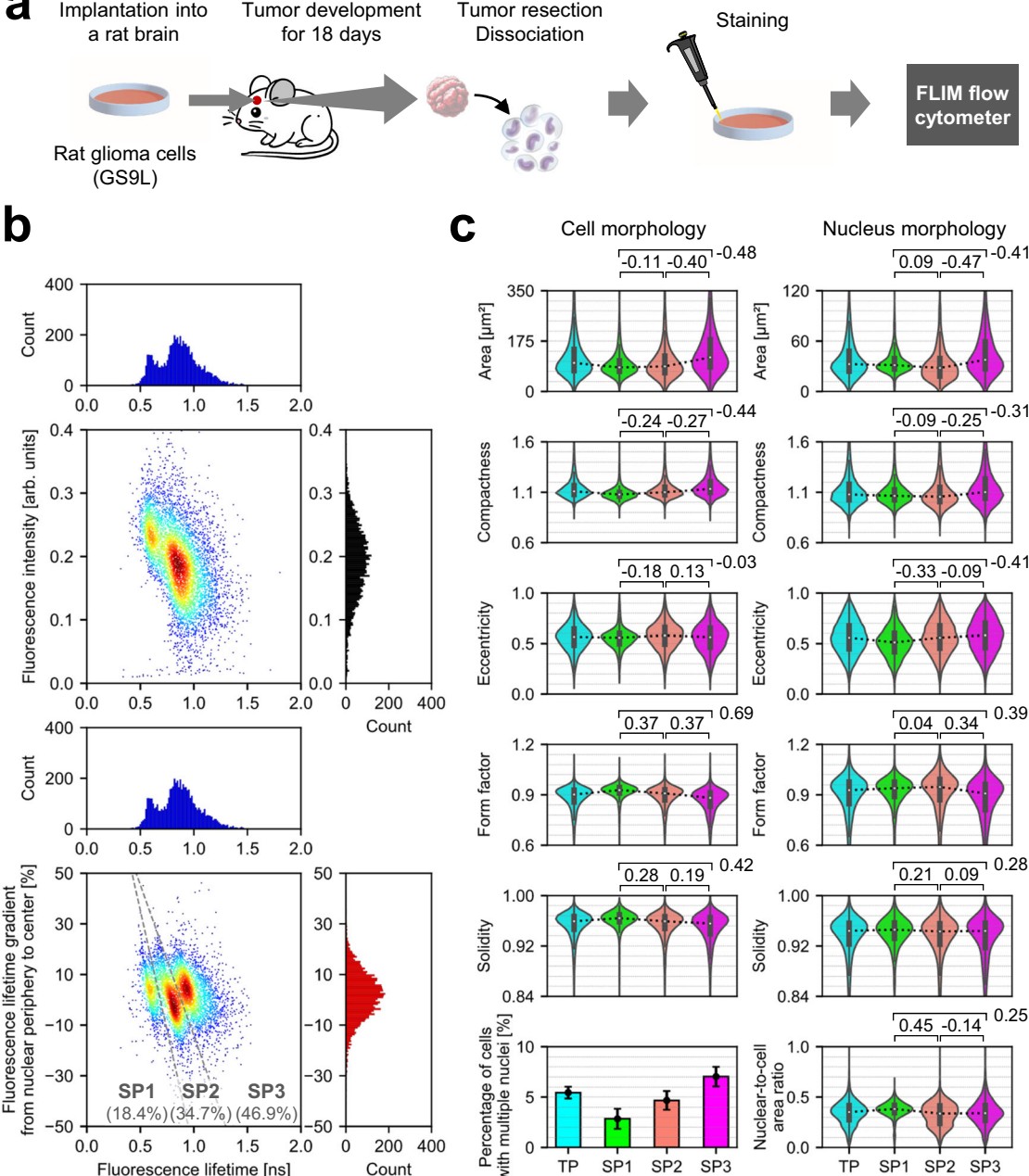

**Fig. 5 | Investigation of cellular heterogeneity in rat glioma with FLIM flow cytometry. a** Experimental procedure. **b** Distributions of tumor-derived rat glioma cells in fluorescence intensity (black histogram), fluorescence lifetime (blue histogram), and fluorescence lifetime gradient from nuclear periphery to center (red histogram). SP: subpopulation. $n = 5732$. **c** Comparison of subpopulations in morphological features of cells and their nuclei. TP: total population. Values between two samples represent effect sizes (Cohen's $d$), with their standard errors <0.04.

The violin plots display the median values with white dots, the first and third quartiles with box edges, and 1.5 times the IQR with whiskers. For the violin plots, the sample sizes are as follows: TP: $n = 5732$; SP1: $n = 1054$; SP2: $n = 1991$; SP3: $n = 2687$. The bar plot illustrates the percentage of cells with multiple nuclei calculated for each cell population, accompanied by the upper and lower bounds of its 95% confidence interval. Source data are provided as a Source Data file.

structural alterations of DNA and chromatin, prior to overall changes in cell and nucleus areas (Supplementary Fig. 11). Consequently, DNA-bound doxorubicin effectively quenched SYTO16 fluorescence[42], particularly pronounced at the center of the nuclei compared to their peripheries, as indicated by the shift in fluorescence lifetime gradient from 5.7% to −6.8% (median) during the 0–140 min period (Fig. 6b). This could be because gene-rich, thus relatively loose chromatin structure located at the central area of nucleus[43,44], which is also indicated by the reduced self-quenching[45] (i.e., longer fluorescence lifetime) at the central area (Supplementary

Fig. 12), was more susceptible to doxorubicin-induced changes compared to the nucleus periphery, gene-poor and condensed area. DNA condensation and loosening may have led to changes in the refractive index around SYTO16, affecting its fluorescence lifetime[46]. Intranuclear structure and its drug-induced change hidden behind fluorescence intensity variations could be visualized by standardized fluorescence lifetime images (Fig. 6c–e). These results showcase the ability of FLIM flow cytometry to detect large-scale temporal intranuclear dynamics that were difficult to capture with flow cytometry and fluorescence lifetime flow cytometry.

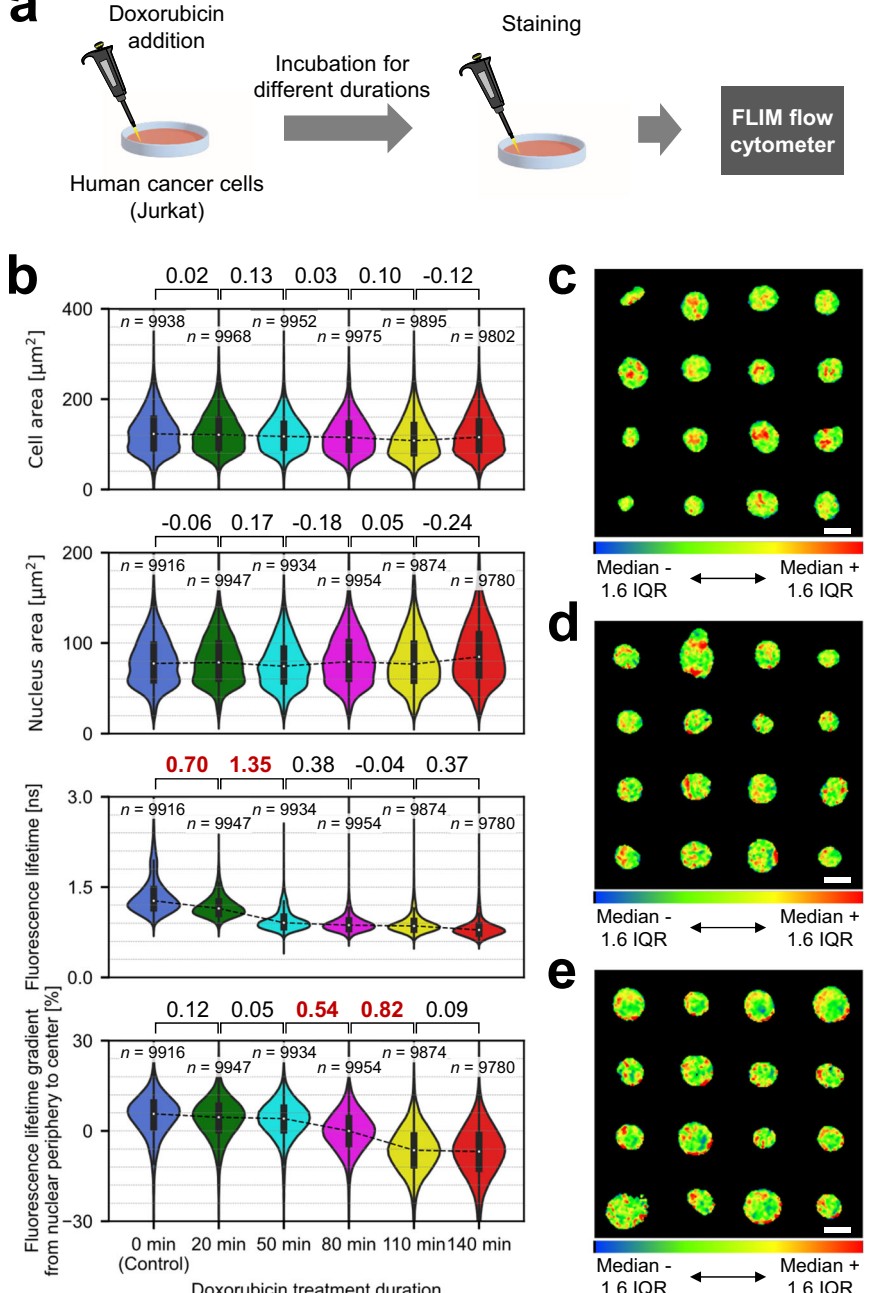

**Fig. 6 | Large-scale analysis of drug-induced temporal nucleus dynamics with FLIM flow cytometry. a** Experimental procedure. **b** Comparison of temporal evolution data in cell area, nucleus area, fluorescence lifetime, and fluorescence lifetime gradient from nuclear periphery to center. Values between two samples represent effect sizes (Cohen's *d*), with their standard errors <0.02. Effect sizes with absolute values exceeding 0.5 are highlighted in bold and red. The violin plots display the median values with white dots, the first and third quartiles with box edges, and 1.5 times the IQR with whiskers. **c-e**, Examples of standardized fluorescence lifetime images of Jurkat cells treated with doxorubicin for 0 min (**c**), 80 min

(**d**), and 140 min (**e**), and stained with SYTO16. The standardization was performed by adjusting the color scale to the range of median ±1.6IQR of fluorescence lifetime pixel values from each nucleus, after compensating for the fluorescence lifetime slope across a field of view and applying a 3 × 3 median filter. Each image set shows the following fluorescence lifetime gradient from nuclear periphery to center (**c**: 12%, **d**: 0%, **e**: −12%, to a tolerance of ±0.1%). Scale bars: 10 µm. Two independent experiments were performed, resulting in similar results. Source data are provided as a Source Data file.

## Discussion

In summary, we experimentally demonstrated high-throughput FLIM flow cytometry at over 10,000 eps (flow speed: ≥3 m/s) based on the high-speed FLIM system with a continuous-wave laser modulated at <222 MHz. Through large-scale fluorescence lifetime image analysis, we detected the hidden subpopulation in male rat glioma and probed drug-induced temporal intranuclear structural changes that, to the

best of our knowledge, were previously undetectable at such high speeds. These results showcase that FLIM flow cytometry holds significant promise for gaining valuable insights across diverse biomedical applications including the evaluation of drug efficacy.

Our FLIM flow cytometry approach effectively mitigates a limitation arising from the finite number of modulation frequencies available for estimating fluorescence lifetime while facilitating high-speed FLIM

image acquisition within a limited bandwidth. Distributing excitation beam spots with varied modulation frequencies along a line allows for simultaneously measuring multiple fluorescence lifetime pixels of flowing objects, similar to frequency-division multiplexing in telecommunications[47]. However, in cases where fluorescence exhibits a multi-exponential decay curve, the use of only a single beam array introduces disparities in the estimated fluorescence lifetime across a field of view (Supplementary Fig. 13a, b). To address this issue, we utilized complementary frequency pairs of intensity modulation from dual beam arrays. Increasing the number of beam arrays from one to two not only enhances the signal-to-noise ratio of fluorescence imaging but also enables us to numerically compensate for the modulation-frequency-dependent artifacts of fluorescence lifetime imaging without expanding the required bandwidth (Supplementary Fig. 13c,d).

We employed modulation frequencies ranging from 21 to 222 MHz for the excitation beam arrays, which necessitates target fluorescence lifetimes in the range of a few nanoseconds. This requirement is attributed to the response function of fluorophores (Eq. 1). When measuring longer lifetimes (e.g., >10 ns)[48], not only do the amplitudes of emitted light signals significantly diminish but also the phases approach $\pi/2$. Consequently, even minor phase differences result in substantial variations in fluorescence lifetimes. Targeting these longer fluorescence lifetimes requires the use of a lower modulation frequency range (<20 MHz), which reduces the effective bandwidth and, in turn, lowers the event rate of the FLIM flow cytometer. This limitation is fundamental to all fluorescence lifetime measurements, not limited to our method.

It is important to note that the absolute fluorescence lifetime value calculated by our method can differ from the value measured by other methods[49,50], such as time-correlated single photon counting (TCSPC)[24], particularly in cases where fluorescence exhibits a multi-exponential decay curve. This is not uncommon for frequency-domain FLIM that employs a single or a limited number of modulation frequencies for exciting fluorophores at a single spatial point[17]. In our FLIM, we solely utilized phases of fluorescence signals (i.e., phase lifetime) and did not consider their modulation depths relative to their averages (i.e., modulation lifetime) when estimating fluorescence lifetimes, as it is not feasible due to the loss of average intensity information for each beam spot by single-pixel photodetection. This approach can result in underestimating fluorescence lifetime values compared with their nominal values (Fig. 3). In fact, we found that the fluorescence lifetimes of SYTO16 observed with our FLIM were shorter than TCSPC-based confocal FLIM (Supplementary Figs. 12 and 14). While this can pose challenges in investigating actual fluorescence lifetime values and their composition ratios for samples, such as static chemical substances and biological materials, instruments with high-speed or high-throughput capability are often unnecessary for this purpose. The advantage of our method lies in achieving high-speed and high-throughput imaging capability, free from fluorescence intensity fluctuations, while efficiently eliminating modulation-frequency-dependent artifacts across an entire field of view (Supplementary Fig. 13), accomplished through a unique pair of intensity-modulated beam arrays.

While we introduced the parameters of fluorescence intensity and fluorescence lifetime gradients for large-scale image-based profiling of heterogeneous cells in this study (Supplementary Fig. 6), it is worth noting that these parameters may not be universally applicable across all imaging methods. As shown in Supplementary Figs. 12 and 14, TCSPC-based confocal FLIM revealed dim regions within the nuclei stained with SYTO16 due to the depth sectioning capability of confocal microscopy, a phenomenon less apparent with our FLIM (Supplementary Figs. 5 and 9). Moreover, fluorescence lifetime values were different between both methods, as mentioned above. These indicate the challenges of applying the same parameters for image analysis to images obtained with different methods.

Since conventional confocal microscopy is suited for detailed observation of a limited number of cells, unlike our high-throughput FLIM flow cytometry, alternative parameters may be more suitable for its image analysis.

While we discriminated cell subpopulations in male rat glioma (Fig. 5), expanding our FLIM flow cytometer and conducting further in-depth analyses are crucial to thoroughly categorize these subpopulations into specific cell types and states. Specifically, integrating additional detection channels for immunofluorescence into our FLIM flow cytometer can enhance the resolution and specificity of subpopulation identification. This can be achieved by separating glioma cells and immune cells with the help of CD markers, and further characterizing subtypes of immune cells[3]. Additionally, incorporating a cell sorting system[25] into our FLIM flow cytometer offers another promising avenue for investigating subpopulations. This technology enables us to isolate specific cell populations and, in turn, perform downstream analyses, such as gene expression analysis, for each subpopulation.

## Methods
### Ethics statement
All animal experiments were approved by the Animal Studies Ethics Committee of Tohoku University Graduate School of Medicine and were conducted at a certified laboratory in Tohoku University Graduate School of Medicine (2021MdA-087-02). Humane endpoints were defined as when rats exhibited symptoms such as decreased activity, reduced water intake, decreased food intake, rapid weight loss (more than 20% in a few days), obvious signs of pain, or prominent indicators of central nervous system failure. We strictly adhered to these guidelines.

### Detailed schematic of high-throughput FLIM flow cytometry
The detailed schematic of the high-throughput FLIM flow cytometer is presented in Supplementary Fig. 1. A continuous-wave 488-nm laser (Coherent Genesis CX 488 STM) served as the light source. Two linearly polarized, intensity-modulated beam arrays were produced through the interference of two sets of 50 beams, each deflected by an acousto-optic deflector (AOD, Brimrose TED-300-200-488) in a Mach-Zehnder interferometric configuration. The AODs were driven by multi-tone signals containing 50 frequency components (ranging from 207.12890625 to 307.6171875 MHz and from 293.26171875 to 393.75 MHz, with a spacing of 2.05078125 MHz), generated by an arbitrary waveform generator at 1.2 Giga-samples per second (GS/s) (Signatec PXDAC4800). In one of the optical paths in the interferometer, a laser beam underwent a frequency shift of 207.6171875 MHz using an acousto-optic frequency shifter (Isomet 1250 C) before reaching the AOD. This resulted in the intensity modulation frequency of each beam array spanning from 20.9960937 to 221.972656 MHz, with a 4.1015625 MHz spacing. After the generation of the intensity-modulated beam arrays, a pair of mirrors was employed to spatially invert one of the beam arrays. Subsequently, they were orthogonally polarized and recombined in the same optical path by utilizing a half-wave plate and a polarizing beam splitter. After that, they were separated in the direction orthogonal to the array beam by a Wollaston prism (Thorlabs WPQ10). A dichroic filter reflected most of the light for excitation toward a customized microfluidic chip (Hamamatsu Photonics) to excite fluorescent molecules, whereas the weak transmitted light was used as a baseline for calculating the phase delays of the fluorescence signals. After being expanded by a 4-f relay-lens system to align the beam diameter with the pupil of an objective lens, the excitation beams were focused by the objective lens (Olympus UPLSAPO20X; NA, 0.75) into the microfluidic channel to scan the flowing object whose speed was controlled by syringe pumps (Harvard Apparatus, H70-4500). Transmitted light, epifluorescence light from the two beam arrays, and reference light (not incident on the sample) were all separately detected by Si avalanche photodetectors (Thorlabs

APD430A/M for the detection of transmitted and the reference light and APD430A2/M for the detection of the fluorescence light). The signals from the photodetectors were converted into digital data by a digitizer with a sampling rate of 1.25 GS/s (Spectrum M4i.2212-x8) and processed using a lab-made LabVIEW program (LabVIEW 2016 and LabVIEW 2021). For our system, it is possible to implement a laser with a different wavelength as an alternative excitation source, as long as we carefully redesign the optical system, including components sensitive to wavelengths, such as AODs and a dichroic filter. Implementing spectrally resolved fluorescence lifetime imaging is also possible by adding dichroic filters or gratings to separately detect fluorescence light based on their wavelengths.

## Fluorescence lifetime calibrations

It was necessary to execute fluorescence lifetime calibrations for the FLIM flow cytometer before or after conducting FLIM flow cytometry. This involved measuring a fluorescence lifetime standard sample exhibiting a mono-exponential fluorescence decay. Specifically, we measured 10-μM Rhodamine B in ethanol, a well-known example demonstrating a mono-exponential fluorescence decay with a fluorescence lifetime of around 2.69 ns[51], and recorded the phase differences between reference signals and fluorescence signals. To determine the phase differences that originated from the FLIM flow cytometry setup, we subtracted the phase differences stemming from a fluorescence lifetime of 2.69 ns from the recorded phase differences. These calibration phases were subsequently subtracted from the phase differences between the reference signals and fluorescence signals in FLIM flow cytometry, leading to the generation of phase differences that originated from the fluorescence lifetime of the target sample.

## Signal processing for image reconstruction

Signal processing to reconstruct images from temporal signals from APDs is a key procedure for FLIM flow cytometry, which is outlined in Supplementary Fig. 2. First, we performed Fourier transformation on the temporal signal to obtain its frequency spectrum. Then, we extracted a sub-band spectrum centered on each modulation frequency using the frequency window equivalent to the modulation frequency spacing between neighboring intensity-modulated beam spots. The extracted sub-band spectrum was low-pass filtered by shifting its sub-band to the zero-frequency position and eliminating approximately 10% of its high-frequency components, which can effectively reduce crosstalk noises induced by undesired interference pairs of beam spots while retaining spatial resolution in a typical setting of our system[26]. After that, we performed inverse-Fourier transformation on the spectrum to obtain a complex line profile of a flowing object. To obtain a bright-field image, we took the absolute value of the profile generated from transmitted light signals and concatenated them. After performing phase calibration of the profile generated from fluorescence signals by using the phases of reference signals and the already obtained calibration phases of Rhodamine B in ethanol, we concatenated amplitude and phase line images and generated an amplitude image pair and a phase image pair. Doubling the number of the image pixels by interpolation, we estimated the horizontal and vertical pixel differences between the image pairs obtained from both fluorescence channels by calculating the 2D cross-correlation of the amplitude image pair after one of the amplitude images was flipped. For Fig. 1d–g and Supplementary Figs. 4, and 13, a $3 \times 3$ median filter was used for phase images after the interpolation. Then, we superposed the amplitude and phase image pairs by translation. The superposed amplitude image (i.e., fluorescence intensity image) was obtained as a simple average of the amplitude image pair. On the other hand, the superposed phase image was obtained as phase values at the center modulation frequency (i.e., the average of the complementary modulation frequency pair) by fitting each phase pair to be superposed under the assumption of the existence of two different

fluorescence lifetime components. Lastly, fluorescence lifetime values were calculated based on the $\tau = \tan\theta/2\pi f_{center}$, where $\tau$ is fluorescence lifetime, $\theta$ is the estimated phase, and $f_{center}$ is the center modulation frequency (around 120 MHz). Alternatively, the simple average of two fluorescence lifetime values was calculated if the assumption of the existence of two fluorescence lifetime components was not feasible with the phase pair being superposed.

The fluorescence lifetime images in Fig. 1d, Fig. 2, Fig. 3, and Supplementary Fig. 4 were segmented with a threshold value of $I_{max}/$ 2.5, where $I_{max}$ is the intensity maximum in each fluorescence intensity image. Similarly, the fluorescence lifetime images in Fig. 1e were segmented with a threshold value of $I_{max}/4.5$ while those in Figs. 1f, g, 4b–d, 5, and 6, Supplementary Figs. 5, 9, and 10 were segmented with a threshold value of $I_{max}/3.5$. The original pixel size of the fluorescence lifetime images, without interpolation, is 0.8 μm (in a beam spot alignment direction) and 0.2–0.8 μm (in a flow direction), depending on the flow speed[26,29,30]. For example, the original pixel size and the number of pixels of each image in Fig. 2a are 0.8 μm × 0.7 μm and ~48 × 11 pixels for beads and 0.8 μm × 0.8 μm and ~50 × 22 pixels for cells, in the horizontal and vertical (flow) directions, respectively.

## Jurkat cell preparation

Jurkat cells (E6.1) used in this study were derived from ECACC and purchased through KAC (Cat. No. EC88042803-F0), with authentication by STR profiling. They were routinely cultured in RPMI-1640 (FUJIFILM Wako Pure Chemical Corporation) supplemented with 10% fetal bovine serum (FBS) (MP Biomedicals), 1% penicillin-streptomycin (FUJIFILM Wako Pure Chemical Corporation) at 37 °C in a 5% CO$_2$ atmosphere. To obtain the data of Jurkat cells in Fig. 2 and Supplementary Fig. 3, the cells in the culture medium were centrifuged, suspended in PBS (FUJIFILM Wako Pure Chemical Corporation) at a concentration of 3.2 × 10⁶ cells/ mL, and stained with 5-μM Calcein-AM (Funakoshi, Cat. No. 341-07901) at 37 °C for 55 min in darkness. Then, they were centrifuged again and resuspended in about 600 μL PBS, followed by FLIM flow cytometry. To obtain the data in Supplementary Fig. 11a-e, the cells in the culture medium were centrifuged and suspended in PBS with 0-μM (negative control) and 3.45-μM doxorubicin (FUJIFILM Wako Pure Chemical Corporation, Cat. No. 040-21521) for 250 min. Viability tests with trypan blue after the doxorubicin treatment showed > 90% viability of both samples. 6.90-μM doxorubicin was also used in Supplementary Fig. 11f. Then, they were stained with 4-μM SYTO16 (Thermo Fisher Scientific, Cat. No. S7578) at 37 °C for 65 min in darkness, centrifuged, and resuspended in PBS, followed by analysis with FLIM flow cytometry. To obtain the data shown in Supplementary Fig. 12, the cells were stained with 4-μM SYTO16 in PBS at 37 °C for 1 hour in darkness, followed by being centrifuged and resuspended in PBS. To obtain the data shown in Fig. 6, Supplementary Figs. 9, and 10, the cells in the culture medium were centrifuged and suspended in the FBS-free culture medium. The cell solution (about 0.9 × 10⁶ cells/mL) was divided into six 1 mL samples. Then, the 1 mL FBS-free culture medium with 10-μM doxorubicin was added to five out of the six samples so that the samples were treated with 5-μM doxorubicin for 20 min, 50 min, 80 min, 110 min, and 140 min, respectively. All six samples were centrifuged and suspended in 4-μM SYTO16 FBS-free culture medium at 37 °C for 50 min in darkness at the same time, centrifuged, and resuspended in PBS, followed by FLIM flow cytometry. To avoid potential systematic errors, we measured all six samples in a random order (measurement order: 20 min, 140 min, 80 min, 50 min, 0 min, and 110 min samples). Also, the Jurkat cells used for Supplementary Fig. 11 and Fig. 6 were imaged on completely different days, two months apart. The range of doxorubicin concentrations and incubation times used in this study was determined based on previous research[42,52].

## *Euglena gracilis* cell preparation

*Euglena gracilis* NIES-48 cells were obtained from the Microbial Culture Collection at the National Institute for Environmental Study[53]. They

were routinely cultured in AF-6 culture medium at 25 °C while being illuminated in a 14:10-hour light: dark pattern. To obtain the data in Fig. 4e. *gracilis* cells were stained with 2.5-μM SYTO16 for 50 min, followed by FLIM flow cytometry.

## Rat glioma cell preparation

Glioma-established rat glioma cell line GS-9L was purchased from ECACC and was not authenticated. The GS-9L cells were maintained in Dulbecco's minimum essential medium (DMEM) (Wako) with 10% FBS and penicillin-streptomycin (100 U/mL, Nacalai). The cells were seeded into a tissue culture dish (Invitrogen) 2 or 3 days before implantation into a rat brain. They were cultured in an incubator at 37 °C in a humidified atmosphere composed of 95% air and 5% $CO_2$. Cells were harvested by trypsinization, washed once with DMEM, and resuspended in PBS (Nacalai) for implantation.

Ten-week-old male Fisher 344 (F344/NSlc) rats (Kumagai-Shigeyasu Co., Ltd.), each weighing between 200 and 250 g, were housed in standard facilities and provided free access to water and food. Sex differences were not considered in this study, as its aim was the proof-of-principle demonstration of the developed FLIM flow cytometer. For intracranial tumor implantation, rats were anesthetized with 2% isoflurane in a mixture of 30% oxygen and 70% nitrous oxide. During surgery, the rectal temperature was maintained at 37 °C ± 0.5 °C using a feedback-regulated heating pad (Bio Research Center, BWT-100). They were then placed in a small-animal stereotaxic frame (Kopf). A sagittal incision was made through the skin, and a burr hole was created in the skull by using a twist drill. The cannula coordinates were 0.5 mm anterior to the bregma and 3 mm lateral from the midline. A cell suspension ($5 \times 10^4$ cells/2 μL) was used for tumor implantation. This suspension was injected at a depth of 4.5 mm from the brain surface. The needle was then removed, and the wound was closed using a nylon suture. For tumor extraction, the rats were anesthetized by aspirating excess isoflurane 18 days after tumor implantation. They were transcardially perfused with saline to remove blood. Tumors were extracted from the rat brain. The tumor tissues were cut into pieces approximately 5 mm in diameter and dissected into single cells by using a gentleMACS™ Dissociator (Milteyi Biotec). The single-cell suspensions were frozen and stocked with a cell freezing buffer (CELLBANKER 1 plus, Takara Bio) to transport from Miyagi to Tokyo in a frozen state along with frozen cultured GS-9L cells. After being defrosted, the tumor-derived rat glioma cells were incubated in RPMI-1640 supplemented with 10% FBS and 1% penicillin-streptomycin at 37 °C in a 5% $CO_2$ atmosphere for 0 min (Supplementary Fig. 14) and for 7 hours and 40 min (Fig. 5 and Supplementary Figs. 5 and 8). The cells used for Supplementary Fig. 14 and for Fig. 5 were from different samples. They were stained with 2.5-μM SYTO16 for 1 hour, followed by analysis with FLIM flow cytometry (Fig. 5 and Supplementary Figs. 5 and 8) or confocal FLIM analysis (Supplementary Fig. 14). On the other hand, cultured GS-9L were stained with 3-μM SYTO16 for 1 hour after being thawed, followed by analysis with FLIM flow cytometry (Supplementary Fig. 7). Also, the defrosting protocol was as follows; we warmed -1 mL of the frozen cell solution with 37 °C water bath for 1–2 min, diluted it with 10 mL of 37 °C pre-warmed RPMI-1640 supplemented 10% FBS and 1% penicillin-streptomycin, centrifuged the solution, removed the supernatant, and resuspended the cells in the RPMI-1640 solution.

## Image analysis

Image analysis for FLIM flow cytometry was performed by using Cell-Profiler 4.2.1[37]. The bright-field, fluorescence intensity, and fluorescence lifetime images obtained with the FLIM flow cytometer were respectively aligned (e.g., 100 × 100 image tiles) and output as combined images that were input into CellProfiler. Segmentation of the target objects in Figs. 2 and 3, and Supplementary Fig. 3 was based on fluorescence intensity images while, in Fig. 4, it was based on both

bright-field and fluorescence intensity images. In Figs. 5 and 6, Supplementary Figs. 7, 8, and 11, the segmentation of cells and their nuclei was based on the bright-field images and fluorescence intensity images, respectively. Here, the nuclei were segmented by using a cut-off value of $I_{max}/3.5$, where $I_{max}$ is the maximum fluorescence intensity for each image tile, and subsequent morphological operations such as dilation, erosion, and closing. After the segmentation, morphological features, intensities (i.e., transmitted light intensity, fluorescence intensity, and fluorescence lifetime), and intensity distributions were extracted. When more than two nuclei were segmented within a cell, the one with the smaller eccentricity was chosen for subsequent analyses of nucleus morphology. Despite a limited number of instances, when aggregated multiple cells were segmented as a single object by CellProfiler, we treated them as single cells with multiple nuclei. Conversely, when multiple nuclei were segmented as a single object inside a cell, we treated it as a single cell with a single nucleus. To calculate fluorescence intensity and fluorescence lifetime in Figs. 2, 3, 5, and 6, Supplementary Figs. 4, 7, 8, and 11, we simply averaged pixel values within the segmented objects. Regarding the fluorescence lifetime gradient and fluorescence intensity gradient from nuclear periphery to center in Figs. 5 and 6, and Supplementary Figs. 6, 7, 8, and 11, we divided the single nucleus into 4 ring-shaped regions by using the MeasureObjectIntensityDistribution module of CellProfiler, calculated fractional fluorescence lifetime and fluorescence intensity in each contour, performed linear fitting with these 4 fractional values, and determined the value changes from the outermost contour (i.e., peripheral area) to the innermost contour (i.e., central area) using the fitted line (Supplementary Fig. 6a). Morphological features such as size, compactness, and eccentricity were extracted by using CellProfiler in Figs. 4–6 and Supplementary Figs. 7, 8, and 11.

## Event rate calculation of high-throughput FLIM flow cytometry

In Fig. 2, we conducted the acquisition of 10,000 consecutively triggered events and calculated the event rate, defined as the number of images containing recognizable object(s) acquired per second. The criteria of recognizable objects were an eccentricity of <0.6 for beads and an eccentricity of <0.72 for Jurkat cells (Supplementary Fig. 3). Fewer than one object was recognized per image, even when multiple objects were imaged in a single image acquisition. Consequently, the actual event rates were always smaller than the average values as shown in the line graphs of Fig. 2b. In addition, along with the event rates, we calculated the digital data generation speed of our FLIM flow cytometer[31]. For beads in Fig. 2, the demonstrated event rate of 11,297 eps corresponds to a digital data generation speed of 108.3 MB/s, which was calculated as follows: 11,297 event/sec × 3352 Byte/event × 3 channels (i.e., bright-field, fluorescence, fluorescence lifetime images) since an effective event acquisition time was 2.6819 μs [11 pixels (the number of pixels of a bead image in a flow direction) divided by 4.1015625 MHz (the line acquisition rate)], which corresponds to 3352 Byte/event using the digitizer with a sampling rate of 1.25 GHz and 8-bit resolution. Similarly, the demonstrated event rate of 10,371 eps for cells in Fig. 2 corresponds to 198.9 MB/s.

## Statistics and reproducibility

Python 3.10.9 was used for statistical analysis. In Figs. 5 and 6, Supplementary Figs. 7, 11, and 14, the effect sizes (Cohen's d) between the $k$th and $i$th samples were calculated, where $k < i$, as follows[54]:

$$d_{k,i} = \frac{M_k - M_i}{\sqrt{\frac{(n_k-1)\text{SD}_k^2 + (n_i-1)\text{SD}_i^2}{n_k + n_i - 2}}}, \tag{2}$$

where $M_i$ ($M_k$), $\text{SD}_i$ ($\text{SD}_k$), and $n_i$ ($n_k$) represent the mean, standard deviation, and sample size of the $i$th ($k$th) sample, respectively. In

addition, standard errors of the effect sizes were calculated as

$$se_{k,i} = \sqrt{\left(\frac{n_k + n_i - 1}{n_k + n_i - 3}\right)\left[\left(\frac{4}{n_k + n_i}\right)\left(1 + \frac{d_{k,i}^2}{8}\right)\right]}. \qquad (3)$$

In Supplementary Fig. 14c, p values were calculated with Welch's t-test (two-sided).

The number of images used in this study per sample typically ranged from 5000 to 10,000 to ensure statistical significance comparable to the standard practice in flow cytometry analyses (see each figure for the precise sample size for each experiment). Objects that were not identified by CellProfiler in the images or that returned NaN (Not a Number) values for their morphological features were excluded from the subsequent analysis to prevent the inclusion of noise. Additionally, while a small number of outliers were not shown in the figures to ensure better visibility of the data distributions, all data points are provided in the Source Data. To avoid potential systematic errors during data collection by the FLIM flow cytometer, the measurement order of the multiple samples was randomized where applicable (e.g., Fig. 6). The investigators were not blinded to allocation during experiments and outcome assessment because a single investigator conducted both the measurements and the analysis.

### FLIM flow cytometry of cultured rat glioma cells (GS-9L)

We measured cultured GS-9L cells, intended for implantation into a rat brain, with FLIM flow cytometry, allowing for comparison with tumor-derived glioma cells shown in Fig. 5. As depicted in Supplementary Fig. 7a, distinct subpopulations were not observed in the cultured cells in any parameters used in Fig. 5b. This indicates that the discernible higher-order phenotypic differences observed in the tumor-derived cells may result from interactions with various tumor microenvironments. The differences in the percentage of cells with multiple nuclei and in the variation of the nuclear-to-cell area ratio between the tumor-derived and cultured cells (Supplementary Fig. 7b) further support this notion. Additionally, the cultured cells exhibited longer fluorescence lifetimes compared to the tumor-derived cells. This disparity could be partly attributed to the larger cell and nucleus areas in the cultured cells (Supplementary Fig. 7b), potentially resulting in diminished self-quenching for the fluorescent molecules, in addition to the smaller cellular heterogeneity in a cultured condition.

### FLIM flow cytometry of Jurkat cells treated with doxorubicin

To further investigate the nucleus changes induced by doxorubicin, we performed FLIM flow cytometry of Jurkat cells treated with doxorubicin for 250 min, which was much longer than the duration used in Fig. 6, and calculated the fluorescence lifetime, fluorescence intensity, and transmitted light intensity for each nucleus (Supplementary Fig. 11a, b). Consequently, the distribution of control Jurkat cells exhibited two characteristic subpopulations, suggesting different nucleus states. In addition, higher fluorescence intensity corresponded to a shorter fluorescence lifetime due to the self-quenching of SYTO16[45]. Moreover, the subpopulation with higher fluorescence intensity (i.e., shorter fluorescence lifetime) exhibited larger cell areas compared to the one with lower fluorescence intensity (i.e., longer fluorescence lifetime) (Supplementary Fig. 11c). After we treated the cells with 3.45-µM doxorubicin for 250 min, the two peaks in fluorescence lifetime became shorter and merged toward a single peak although the peaks in fluorescence intensity and transmitted intensity remained partly overlapped (Supplementary Fig. 11b). This indicates that the surrounding environment of the fluorophores (i.e., SYTO16) expressed by their fluorescence lifetime was varied toward a single state by the doxorubicin treatment, regardless of DNA content. The potential cause of this change may be non-radiative energy transfer between SYTO16 and doxorubicin molecules[42] supported by

chromatin condensation[52], which is known to be induced by doxorubicin. Meanwhile, it is understandable that since the fluorescence intensity and transmitted light intensity are mainly determined by the amount of the DNA content, the relative positions of the two peaks of their values showed no obvious change before and after the doxorubicin treatment. However, the decreased transmitted light intensity after the doxorubicin treatment may indicate the effect of chromatin condensation or other nucleus structure alterations that contribute to the increased scattering of intensity-modulated light by the nucleus.

Next, we statistically investigated the intranuclear fluorescence lifetime distribution of the Jurkat cells before and after the doxorubicin treatment. Specifically, we divided the area of each nucleus into four ring-shaped regions (i.e., contours) and measured fractional values in each contour with respect to the entire nucleus in the same manner as in calculating fluorescence lifetime gradient (Supplementary Fig. 6). As shown in Supplementary Fig. 11d, the variations in fluorescence lifetime distributions among intranuclear areas were smaller compared to those in fluorescence intensity, indicating the robustness of fluorescence lifetime. Moreover, the fractional fluorescence lifetime values decreased from the center of the nucleus to its periphery. This trend in intranuclear fluorescence lifetime distribution could be explained by the relatively loose and sparse chromatin structure at the central nucleus area[43], attributed to frequent gene transcription, leading to the reduced self-quenching of SYTO16. On the other hand, as shown in Supplementary Fig. 11e, the trend in fluorescence lifetime distribution reversed (i.e., the central areas of the nuclei showed shorter fluorescence lifetimes) after doxorubicin treatment while fluorescence intensity distribution remained largely unchanged. This reversal could be explained by chromatin condensation or other intranuclear structural alterations induced by doxorubicin, leading to changes in the intranuclear distributions of non-radiative energy transfer efficiency between SYTO16 and doxorubicin[42].

Furthermore, we calculated the same parameters used in Fig. 6b for control cells and those treated with 3.45-µM and 6.90-µM doxorubicin for 250 min (Supplementary Fig. 11e). In comparison to the results in Fig. 6b (0–140 min doxorubicin treatment), we observed more pronounced changes in cell and nucleus areas for the 250 min doxorubicin treatment while similar changes in fluorescence lifetime and fluorescence lifetime gradient from nuclear periphery to center. This supports the notion, as suggested in Fig. 6b, that the change in fluorescence lifetime precedes the change in cell and nucleus areas.

### Validation with TCSPC-based confocal FLIM

To validate some of our findings with conventional FLIM, we conducted experiments using a TCS SP8-FALCON confocal laser-scanning microscope (Leica Microsystems, Wetzeler, Germany) equipped with a pulsed diode laser (PDL 800-B, 470 nm; PicoQuant, Berlin, Germany) operating at a repetition rate of 20 MHz. The emitted fluorescence from 500 to 640 nm was obtained through an HC PL APO 63×/1.40 Oil CS2 objective (Leica Microsystems). We generated a fluorescence lifetime image for Supplementary Fig. 12a in a 256×256-pixel format with 100 repetitive measurements, while we generated images for Supplementary Fig. 14a in a 512×512-pixel format with 15 repeat measurements. To support the findings in Fig. 6, we obtained a fluorescence lifetime image of Jurkat cells stained with SYTO16. The TCSPC-based confocal FLIM revealed not only internuclear but also intranuclear differences in fluorescence lifetime (Supplementary Fig. 12a,b). In particular, the dim regions near the center of the nuclei showed longer fluorescence lifetimes, as evidenced by the negative correlations between fluorescence intensity and fluorescence lifetime pixel values (Supplementary Fig. 12c), which is consistent with our findings in Fig. 6b. To support the findings in Fig. 5, we obtained fluorescence lifetime images of SYTO16-stained cells derived from distinct male rat glioma. The TCSPC-based confocal FLIM revealed clear cell-cell differences[53] in fluorescence lifetime (Supplementary

Fig. 14a). Furthermore, by applying a gate of a fluorescence lifetime of 7 ns to segregate nuclei with shorter and longer lifetimes (Supplementary Fig. 14b) and analyzing their morphological features, we found consistency in nucleus morphology with our findings in Fig. 5c (Supplementary Fig. 14c).

To analyze images obtained with the TCSPC-based confocal FLIM, we used Python 3.10.7 and the OpenCV 4.5.1.48 library. Initially, we converted fluorescence intensity images to grayscale and applied a median filter (cv2.medianBlur, ksize = 3) to eliminate noise. Next, we calculated an adaptive threshold using the Gaussian method to binarize the images, aiming to segment the nuclei from the background. Then, we applied morphological closing twice (cv2.MORPH_CLOSE, ksize1 = 5, ksize2 = 3) to fill in abnormal nuclear pixels and used a median filter (cv2.medianBlur, ksize = 7) once more to filter out background noise generated during processing. After that, we utilized the Laplacian edge detection function to compute the second-order gradient of the images, obtaining boundary information for each nucleus. The edge information was then converted into contours using the contour detection function. These extracted contours serve as a mask that can be used for contour extraction of fluorescence lifetime images (Supplementary Fig. 12b). Finally, using morphology calculation functions such as cv2.contourArea and cv2.arcLength, we obtained morphological parameters of the nucleus.

### Sensitivity

Our FLIM offers unique sensitivity compared to conventional laser-scanning imaging techniques. This is due to the multiplexing of image pixel acquisition in our FLIM, which allows for an increase in pixel dwell time, and thus signal strength per pixel by a factor of $N_{spots}$, the number of beam spots, compared to conventional techniques with the same line-scan rate[26]. However, the superposition of shot noises from all beam spots in our FLIM leads to a decrease in the signal-to-noise ratio (SNR) by a factor of $\sqrt{N_{flu}}$, where $N_{flu}$ represents the number of beam spots used for fluorescence excitation per line scan, assuming a uniform fluorophore distribution. As a result, the SNR improvement can be increased by a factor of $\sqrt{N_{spots}}/\sqrt{N_{flu}}$.

### Reporting summary

Further information on research design is available in the Nature Portfolio Reporting Summary linked to this article.

## Data availability

The images and other associated raw data generated in this study have been deposited in the Figshare database [https://doi.org/10.6084/m9.figshare.26403916] and [https://doi.org/10.6084/m9.figshare.26403919]. Source data are provided with this paper.

## Code availability

The codes used in this study for image analysis and data analysis are available in the Figshare database [https://doi.org/10.6084/m9.figshare.26403916].

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

## Acknowledgements
This work was supported by JSPS Core-to-Core Program (K.G.), JSPS KAKENHI Grant Numbers 19H05633 and 20H00317 (K.G.), Ogasawara Foundation (K.G.), Nakatani Foundation (K.G.), Konica Minolta Foundation (K.G.), Humboldt Foundation's Philipp Franz von Siebold Award (K.G.), Humboldt Association of Japan (K.G.), Precise Measurement Technology Promotion Foundation (H.M.), Konica Minolta Imaging Science Award (H.M.), JST PRESTO (JPMJPR1878) (K.H.), JST FOREST (21470594) (K.H.), JSPS Grant-in-Aid for Scientific Research (B) (23K23297) (K.H.), JSPS Grant-in-Aid for Young Scientists (20K15227) (K.H.), Research Foundation for Opto-Science and Technology (K.H.), JSPS KAKENHI Grant Numbers 21J10600 and 24K18149 (H.K.), Konica Minolta Light Future Incentive Award (H.K.), and JSPS KAKENHI Grant Number 23K27432 (S.R.). We thank Mayu Sehara for her help with the cell sample preparation. The manuscript underwent editing with the assistance of a large language model (LLM).

## Author contributions
K.G. conceived the idea. H.K. and H.M. designed and built the FLIM flow cytometer. H.K. and K.H. validated the system. H.K. developed the programs for FLIM flow cytometry, conducted the pre-imaging sample preparations, carried out the imaging experiments, and analyzed the resulting image data. At. N. and Ar. N prepared rat glioma cells. H.K., K.O., and L.F. conducted validation experiments with the commercially available confocal FLIM system. H.K., K.H., H.M., At.N., S.Y., Ar.N., S.R., K.N, and K.G. interpreted the data. K.H., H.M., and K.G. supervised the optics-related work. S.R. and K.N. supervised the glioma-related work. K.G. obtained funding for the work. All the authors intellectually contributed to the paper and participated in writing the paper.

## Competing interests
K.G. is a shareholder of CYBO. M.S. and K.G. are shareholders of Fly-Works. H.K., K.H., H.M., and K.G. have filed a patent application covering high-throughput fluorescence lifetime imaging flow cytometry. The other authors have no competing interests.
