## [Peer Review File · Nature Communications]

Reviewers' Comments:

Reviewer #1:

Remarks to the Author:

Reviewer's comments on manuscript „FLIM flow cytometry reveals cellular characteristics hidden behind fluorescence intensity fluctuations (#NCOMMS-24-01072)“ by H. Kanno et al.

The manuscript is about realizing fluorescence lifetime imaging of moving cells in a microfluidic flow chamber at an unprecedentedly high speed and demonstrating the capabilities of the new technique. The aims of the manuscript are both important and interesting. On the one side technical development of flow cytometry is important because this is a vital research tool in molecular biology and medical biology. Otherwise the interest lies in that both the realization of fluorescence lifetime measurement on the one side and realizing imaging on the other, in flow conditions, are already challenging individually, notwithstanding the combination of the two. This is a well written manuscript both formally and contentually. Besides the technical development, new analysis features made available by imaging, is the correlative treatment with lifetime of lot of morphological characteristics and spatial gradients of lifetime and fluorescence intensity. I hotly recommend its publication, because this is great leap-forward in the field of flow cytometry, with relevance also to the conventional „resting microscopy“, as to the new analysis insights. My notices, questions and suggestions, listed below are related mainly to the details, and to the better understanding of the text.

Subjectal concerns:

1. Only one side of the phase-modulation technique is treated here, namely the phase lifetime. This is understandable, because of the relative easyness of phase measurements. The title speaks about „intensity fluctuations“, and these fluctuations, or heterogeneities are reflected back by the deviations of the modulation (demodulation) and phase lifetimes. Generally, the phase and modulation lifetimes (ingredients also of the AB-plots) serve as the lower and upper limits of the lifetime distributions, with the average of the phase and modulation lifetimes supplying the mean lifetime of the distribution. My suggestion is the computation of also the modulation lifetime, if it feasible at all, and comparing to the phase lifetimes to reveal possible heterogenities in some of the considered biological examples. Or at least presenting a short discussion on the question of feasibility, or non-feasibility of modulation lifetime in the text.
2. On Page 5, in Rows 155 and 156, the authors state that on beads the measured lifetimes underestimate the „official“ lifetimes. Could it be attributed to the above mentioned lifetime heterogeneities of beads? Namely, to that considering the phase lifetime alone, it under-estimates the mean lifetime.
3. An anomaly related to homo-FRET is treated in several places in the text (e.g. in Rows 217-218 on Page 7, Rows 449-451 and Rows 458-460 on Page 13). Namely that with increasing lifetime, fluorescence intensity reduces. This statement in the text should be refined. Homo-FRET occuring alone can not change fluorescence lifetime because both the primarily and indirectly excited fluorophores are detected simultaneously. Lifetime shortening is due to the quenching by the inevitably present dim (or dark) complexes acting as energy traps. (Presence of homo-FRET can also be suspected from both the phase and modulation lifetimes as „tangent defect“, revealed also by „AB-plots“).
4. The „measure of interaction“ used here, Cohen's d defined in Eqs. 2-3 is seems to be a component also of the more popular Student's two-sample t-test formula. Is this a coincidence only, or there is some real connection between the Cohen's d and Student's (Gosset's) t? It might be described also in the text.
5. Measured lifetimes can be compared not only to other standard measured values, but also to

values suggested by the Strickler-Berg formula. Lifetime heterogeneities could also be attributed to refractive index heterogeneities, inferred from e.g. bright field images. Condensation, loosening effects of cell nucleus coupled to mitotic phases of cells can result in refractive index changes and as a consequence to lifetime changes.

6. Crucial point is the production of the pair of columns of 50 beamlets for excitation. The „interference of multiple beams deflected from the acousto-optic deflectors” (in Rows 265, 266 on Page 8) is rather obscure. E.g. how many beams from the deflectors should interfere to produce the 50 beamlets? An illustrating figure in the Supporting Information is recommended.

Formal concerns:

1. Suggested is writing out in details the total name „fluorescence lifetime imaging (microscopy) [FLI(M)]” in the manuscript title, because the imaging part remains largely suppressed otherwise (with only FLIM).
2. For what is standing for the GS/s unit, e.g. in Rows 269, 285 on Page 8? Giga-cycle/sec? It should be written out in the text.
3. Are all the 7-8 decimal fractional digits necessary? Or the first 2-3 are already adequate, in Rows 267-272 on Page 8.

Reviewer #2:

Remarks to the Author:

In this paper, Kanno et al. present a high-speed and high-throughput fluorescence lifetime imaging microscopy (FLIM) flow cytometry. This instrument is developed by using dual intensity-modulated beam arrays with complementary modulation frequency pairs for fluorophore excitation. Using this approach, this system can acquire fluorescence lifetime images of rapidly flowing cells. This technique has been used to identify previously indiscernible subpopulations in rat glioma and to capture dynamic changes in the cell nucleus induced by an anti-cancer drug.

In my opinion, this paper can certainly be published in Nature Communications. The technology itself carries great significance. It is developed on the world's fastest FLIM system with an intensity-modulated continuous-wave laser. It demonstrates high-throughput FLIM flow cytometry at over 10,000 eps, which is one order of magnitude higher than the state-of-the-art. The biological applications are meticulously selected to showcase the spatially resolved imaging and the high frame rate of this system. The result interpretation is rather complete. For any discrepancy between the experiment and the ground truth, the authors provided detailed explanations. The figure presentation is outstanding.

I have the following comments for the authors to consider:

1. Introduction (Line 42)

The authors may want to add a few sentences to elaborate on the limitations of intensity-based measurement. In particular, why the absolute intensity value is a meaningful quantity to measure? What biological (or physiological) parameters are linked to the probe's absolute intensity? Otherwise, people could argue that the use of normalized intensity for visualization is sufficient, and the absolute value becomes irrelevant. This discussion can better reflect why inaccurate measurement of intensity is a problem, hence better justifying the necessity to use lifetime to overcome this problem.

A related comment is for Fig. 3b. I think it would be better if the authors could use different colors to label different types of beads in the left panel. In the current figure, the authors show that 1 is overlapped with 2, and 5 is overlapped with 3 in intensity. However, we do not know if there are any outliers from the lifetime measurement.

2. Demultiplexing (Line 102)

The author may consider including more details about this process in Supplementary Materials. In particular, how would the authors select the time window? Does the digital low-pass filtering induce any artifacts (e.g., ringing) and a reduction in spatial and/or temporal resolution?

3. Supplementary Fig. 5

The authors show that the lifetime increases from the center to the periphery, which is quite interesting. However, it is not clear whether this increase is due to the biological process that the authors discussed in the manuscript or due to other factors. Although fluorescence lifetime is indeed pretty resilient to environmental factors, it may be influenced by some factors (e.g., reflective index). It would be nice if the authors could show a control experiment. For example, the authors could analyze the bead's data to show an unchanged lifetime in different regions from the center to the periphery. In this way, the interpretation of Supplementary Fig. 5 can rule out some systematic errors.

4. Supplementary Fig. 11

Could the author add an explanation in the manuscript as to why a single modulation function can introduce the artifacts shown in the left panels in Figs. 11 a and b?

Reviewer #3:

Remarks to the Author:

The manuscript "FLIM flow cytometry reveals cellular characteristics hidden behind fluorescence intensity fluctuations" describes new device for flow cytometry combined with FLI detecting capability. Fluorescence Lifetime Imaging (FLI) is a powerful method, bringing more quantitative dimension into conventional fluorescent imaging and microscopy experiments. It can be combined with dyes, nanosensors, fluorescent proteins and cellular autofluorescence to measure variety of parameters, including interaction of fluorescent drugs with cellular DNA, when combined with live nuclear stains, as it is demonstrated in the manuscript with Syto 16 and doxorubicin.

Overall, it is very important to bring FLI into flow cytometry, as with this manuscript. However, flow cytometry by itself is a more destructive method, leading to cell stress, compared with e.g. label-free FLIM of live cells, organoids or live animals. Therefore, I recommend revising the manuscript to address the comments presented below.

My major concern is about the observed finding on discriminating the subpopulations of rat glioma cells - would it be possible to validate this finding with 'conventional' FLIM microscopy?

Apart from this, I have only minor comments:

- page 4, "non-single exponential decay" - did you mean 'non mono-exponential'?
- The device is based on 488 nm laser, could you please comment on the possibility of using other wavelengths and spectrally resolved FLI with your cytometry approach in general?
- What is the general sensitivity of the device, compared to conventional confocal FLIM (microscopy)? How the used concentrations (and long incubation times for staining) of Syto 16, Dox etc. are correlated with those used in FLIM with live cells? What about potential toxicity and effects on e.g. cell nuclei, metabolic activity etc.?
- Please explain 'fluorescence lifetime gradient' (page 6, referring to Supplementary Fig. 5). What do you mean by this? Can you validate it using FLIM?
- In general, calling the method as FLIM flow cytometry is a bit confusing. A lot of researchers mean microscopy by FLIM, and simply imaging by FLI. Some 'widefield-like' images are generated but validating with 'real FLIM' would make the manuscript much more interesting and appealing to broader audience. Please also add some discussion on pros/cons of presented method compared

with FLIM and which assays can be useful on this new cytometry method.

To Reviewer #1:

We are grateful to the Reviewer for taking the time to review our manuscript and give us his/her valuable comments. We have taken all the comments into consideration and have made appropriate changes to the manuscript. Our point-by-point response appears below, in which we first echo the Reviewer's comments (shown in italic) and then respond to them. All the changes to the manuscript and Supplementary Information are highlighted in red.

Reviewer #1's comment #1:

The manuscript is about realizing fluorescence lifetime imaging of moving cells in a microfluidic flow chamber at an unprecedentedly high speed and demonstrating the capabilities of the new technique. The aims of the manuscript are both important and interesting. On the one side technical development of flow cytometry is important because this is a vital research tool in molecular biology and medical biology. Otherwise the interest lies in that both the realization of fluorescence lifetime measurement on the one side and realizing imaging on the other, in flow conditions, are already challenging individually, notwithstanding the combination of the two. This is a well written manuscript both formally and contentually. Besides the technical development, new analysis features made available by imaging, is the correlative treatment with lifetime of lot of morphological characteristics and spatial gradients of lifetime and fluorescence intensity. I hotly recommend its publication, because this is great leap-forward in the field of flow cytometry, with relevance also to the conventional „resting microscopy”, as to the new analysis insights. My notices, questions and suggestions, listed below are related mainly to the details, and to the better understanding of the text.

Authors' response:

We thank the Reviewer for providing the positive comment and acknowledging the significance of our research and manuscript.

Reviewer #1's comment #2:

Subjectal concerns:

1. Only one side of the phase-modulation technique is treated here, namely the phase lifetime. This is understandable, because of the relative easyness of phase measurements. The title speaks about „intensity fluctuations”, and these fluctuations, or heterogeneities are reflected back by the deviations of the modulation (demodulation) and phase lifetimes. Generally, the phase and modulation lifetimes (ingredients also of the AB-plots) serve as the lower and upper limits of the lifetime distributions, with the average of the phase and modulation lifetimes supplying the mean lifetime of the distribution. My suggestion is the computation of also the modulation lifetime, if it feasible at all, and comparing to the phase lifetimes to reveal possible heterogeneities in some of the considered biological examples. Or at least presenting a short discussion on the question of feasibility, or non-feasibility of modulation lifetime in the text.

Authors' response:

We thank the Reviewer for the comment. We concur with the Reviewer's suggestion regarding the potential benefits of measuring modulation lifetimes alongside phase lifetimes for further looking into heterogeneities in the biological examples. However, it is not feasible in our case due to the following reason. Generally, the modulation lifetime, m , is determined by the formula $m = (B/A)/(b/a)$, where a and A denote the average intensities of excitation light and fluorescence, respectively, while b and B represent the amplitudes of intensity-modulated excitation light and

fluorescence, respectively. Since pixel acquisitions of fluorescence lifetimes are multiplexed in our method to enable the world's fastest FLIM, fluorescence signals from all beam spots from a single beam array are simultaneously detected using a single-pixel photodetector. Consequently, the photoelectronic signals containing information of a and A for all the beam spots converge at the direct current (DC) value (i.e., zero frequency position), resulting in their loss, although the information of b and B for the beam spots can be reproduced through Fourier transformation because of their discrete modulation frequencies. To clarify this point, we have added the descriptions regarding the non-feasibility of estimating modulation lifetime to the discussion section on Page 8 (Line 258).

In our FLIM, we solely utilized phases of fluorescence signals (i.e., phase lifetime) and did not consider their modulation depths relative to their averages (i.e., modulation lifetime) when estimating fluorescence lifetimes, as it is not feasible due to the loss of average intensity information for each beam spot by single-pixel photodetection.

Reviewer #1's comment #3:

2. On Page 5, in Rows 155 and 156, the authors state that on beads the measured lifetimes underestimate the „official” lifetimes. Could it be attributed to the above mentioned lifetime heterogeneities of beads? Namely, to that considering the phase lifetime alone, it under-estimates the mean lifetime.

Authors' response:

We thank the Reviewer for the comment. As pointed out by the Reviewer, there should be a discrepancy between phase and modulation lifetimes in cases where fluorescence exhibits a multi-exponential decay curve, which can contribute to underestimating fluorescence lifetime values compared with their nominal values. To clarify this point, we have added the descriptions below to the relevant discussion section on Page 8 (Line 258).

In our FLIM, we solely utilized phases of fluorescence signals (i.e., phase lifetime) and did not consider their modulation depths relative to their averages (i.e., modulation lifetime) when estimating fluorescence lifetimes, as it is not feasible due to the loss of average intensity information for each beam spot by single-pixel photodetection. This approach can result in underestimating fluorescence lifetime values compared with their nominal values (Fig. 3).

Reviewer #1's comment #4:

3. An anomaly related to homo-FRET is treated in several places in the text (e.g. in Rows 217-218 on Page 7, Rows 449-451 and Rows 458-460 on Page 13). Namely that with increasing lifetime, fluorescence intensity reduces. This statement in the text should be refined. Homo-FRET occurring alone can not change fluorescence lifetime because both the primarily and indirectly excited fluorophores are detected simultaneously. Lifetime shortening is due to the quenching by the inevitably present dim (or dark) complexes acting as energy traps. (Presence of homo-FRET can also be suspected from both the phase and modulation lifetimes as „tangent defect”, revealed also by „AB-plots”).

Authors' response:

We thank the Reviewer for the valuable information and suggestion. We reviewed Runnels & Scarlata, *Biophys. J.* **69**, 1569-1583, 1995 ([https://doi.org/10.1016/S0006-3495\(95\)80030-5](https://doi.org/10.1016/S0006-3495(95)80030-5)), and found that the fluorescence lifetime of a group of fluorescent dyes undergoing homo-FRET matches that of a single dye not undergoing homo-FRET. As the Reviewer suggested, the shortening of the fluorescence lifetime is likely to be due to the self-quenching of SYTO16, although the

detail of the mechanism is still elusive. The FRET from the excited SYTO16 to dark or non-fluorescent SYTO16 complex would be one plausible explanation for the shorter fluorescence lifetime. For example, Sparrow & Tippett (*J. Immunol. Methods* **305**, 173-187, 2005, <https://doi.org/10.1016/j.jim.2005.07.017>) reported decreased fluorescence of SYTO16 with nuclear condensation and suggested the self-quenching of SYTO16 for this. We have revised the relevant texts as follows:

In Lines 134-135: The negative correlation between fluorescence lifetime and fluorescence intensity for the cells can be attributed to the **self-quenching of fluorescent molecules**³⁴.

In Lines 218-221: This could be because gene-rich, thus relatively loose chromatin structure located at the central area of nucleus^{43,44}, which is also indicated by the **reduced self-quenching**⁴⁵ (i.e., longer fluorescence lifetime) at the central area (**Supplementary Fig. 12**), was more susceptible to doxorubicin-induced changes compared to the nucleus periphery, gene-poor and condensed area.

In Lines 494-496: This disparity could be partly attributed to the larger cell and nucleus areas in the cultured cells (**Supplementary Fig. 7b**), potentially resulting in diminished **self-quenching** for the fluorescent molecules, in addition to the smaller cellular heterogeneity in a cultured condition.

In Lines 503-504: In addition, higher fluorescence intensity corresponded to a shorter fluorescence lifetime **due to the self-quenching of SYTO16**⁴⁵.

In Lines 525-528: This trend in intranuclear fluorescence lifetime distribution could be explained by the relatively loose and sparse chromatin structure at the central nucleus area⁴², attributed to frequent gene transcription, leading to **the reduced self-quenching of SYTO16**.

Reviewer #1's comment #5:

4. The „measure of interaction” used here, Cohen’s *d* defined in Eqs. 2-3 is seems to be a component also of the more popular Student’s two-sample *t*-test formula. Is this a coincidence only, or there is some real connection between the Cohen’s *d* and Student’s (Gosset’s) *t*? It might be described also in the text.

Authors’ response:

We thank the Reviewer for the comment. The role of the effect size (Cohen’s *d*) is to quantify the standardized difference between the means of two samples. On the other hand, the Student’s *t*-test is designed to assess whether the means of two samples differ with statistical significance or not. Both Cohen’s *d* and Student’s *t* include the mean difference and pooled standard deviation. Consequently, both metrics are related (Cohen, “Statistical Power Analysis for the Behavioral Sciences (2nd ed.),” 1988) but serve different purposes. The reason why the *t*-test was not mainly used in this study is that the *p*-values typically become exceedingly small due to the large sample size inherent in flow cytometry data ($n > 5,000$ for our data), rendering them practically meaningless.

Reviewer #1's comment #6:

5. Measured lifetimes can be compared not only to other standard measured values, but also to values suggested by the Strickler-Berg formula. Lifetime heterogeneities could also be attributed to refractive index heterogeneities, inferred from e.g. bright field images. Condensation, loosening effects of cell nucleus coupled to mitotic phases of cells can result in refractive index changes and as a consequence to lifetime changes.

Authors' response:

We thank the Reviewer for the suggestion. According to Jeong & Oh, *Opt. Express* **27**, 36075-36087, 2019 (<https://doi.org/10.1364/OE.27.036075>), the refractive index of DNA film (0.37 wt% DNA) was reported to be 1.53-1.58, significantly higher than that of water. Thus, DNA condensation and loosening may lead to considerable changes in the refractive index around SYTO16, and in turn, affect the fluorescence lifetime. We agree with the Reviewer. We have added the text below to the Results section on Page 7 (Line 221).

DNA condensation and loosening may have led to changes in the refractive index around SYTO16, affecting its fluorescence lifetime⁴⁶.

Reviewer #1's comment #7:

6. Crucial point is the production of the pair of columns of 50 beamlets for excitation. The „interference of multiple beams deflected from the acousto-optic deflectors” (in Rows 265, 266 on Page 8) is rather obscure. E.g. how many beams from the deflectors should interfere to produce the 50 beamlets? An illustrating figure in the Supporting Information is recommended.

Authors' response:

We thank the Reviewer for the comment. As the Reviewer pointed out, the description of “multiple beams” was unclear. In reality, we generated 50 beamlets for excitation by combining two sets of 50 beams, each deflected by an acousto-optic deflector driven by 50 frequencies. To clarify this point, we have revised the Methods section on Page 9 (Line 293) as follows:

Two linearly polarized, intensity-modulated beam arrays were produced through the interference of **two sets of 50 beams**, each deflected **by an** acousto-optic deflector (AOD, Brimrose TED-300-200-488) in a Mach-Zehnder interferometric configuration.

Additionally, we have slightly modified the left inset in Supplementary Fig. 1 and its caption for the clarity.

Reviewer #1's comment #8:

Formal concerns:

1. Suggested is writing out in details the total name „fluorescence lifetime imaging (microscopy) [FLI(M)]” in the manuscript title, because the imaging part remains largely suppressed otherwise (with only FLIM).

Authors' response:

We thank the Reviewer for the suggestion. We have proposed a new title, “High-throughput fluorescence lifetime imaging flow cytometry,” to emphasize the imaging aspect of FLIM flow cytometry.

Reviewer #1's comment #9:

2. For what is standing for the GS/s unit, e.g. in Rows 269, 285 on Page 8? Giga-cycle/sec? It should be written out in the text.

Authors' response:

We thank the Reviewer for the comment. The unit of GS/s stands for Giga-Sample/sec, which is the general unit for describing the specifications of analog-to-digital and digital-to-analog devices. We have written out the unit for clarity.

Reviewer #1's comment #10:

3. Are all the 7-8 decimal fractional digits necessary? Or the first 2-3 are already adequate, in Rows 267-272 on Page 8.

Authors' response:

We thank the Reviewer for the comment. While including 7-8 decimal fractional digits is unnecessary for the interpretation of our research findings, it can be necessary from an engineering standpoint for reproducing our FLIM flow cytometer using identical electronic devices (i.e., arbitrary waveform generator, digitizer, etc.). Therefore, we chose to include these details in the Methods section.

To Reviewer #2:

We are grateful to the Reviewer for taking the time to review our manuscript and give us his/her valuable comments. We have taken all the comments into consideration and have made appropriate changes to the manuscript. Our point-by-point response appears below, in which we first echo the Reviewer's comments (shown in italic) and then respond to them. All the changes to the manuscript and Supplementary Information are highlighted in red.

Reviewer #2's comment #1:

In this paper, Kanno et al. present a high-speed and high-throughput fluorescence lifetime imaging microscopy (FLIM) flow cytometry. This instrument is developed by using dual intensity-modulated beam arrays with complementary modulation frequency pairs for fluorophore excitation. Using this approach, this system can acquire fluorescence lifetime images of rapidly flowing cells. This technique has been used to identify previously indiscernible subpopulations in rat glioma and to capture dynamic changes in the cell nucleus induced by an anti-cancer drug.

In my opinion, this paper can certainly be published in Nature Communications. The technology itself carries great significance. It is developed on the world's fastest FLIM system with an intensity-modulated continuous-wave laser. It demonstrates high-throughput FLIM flow cytometry at over 10,000 eps, which is one order of magnitude higher than the state-of-the-art. The biological applications are meticulously selected to showcase the spatially resolved imaging and the high frame rate of this system. The result interpretation is rather complete. For any discrepancy between the experiment and the ground truth, the authors provided detailed explanations. The figure presentation is outstanding.

Authors' response:

We thank the Reviewer for giving us the positive comment and acknowledging our work.

Reviewer #2's comment #2:

1. Introduction (Line 42)

The authors may want to add a few sentences to elaborate on the limitations of intensity-based measurement. In particular, why the absolute intensity value is a meaningful quantity to measure? What biological (or physiological) parameters are linked to the probe's absolute intensity? Otherwise, people could argue that the use of normalized intensity for visualization is sufficient, and the absolute value becomes irrelevant. This discussion can better reflect why inaccurate measurement of intensity is a problem, hence better justifying the necessity to use lifetime to overcome this problem.

A related comment is for Fig. 3b. I think it would be better if the authors could use different colors to label different types of beads in the left panel. In the current figure, the authors show that 1 is overlapped with 2, and 5 is overlapped with 3 in intensity. However, we do not know if there are any outliers from the lifetime measurement.

Authors' response:

We thank the Reviewer for the valuable comments and suggestions. We have added a sentence and modified the relevant Introduction section (Page 2) to highlight the limitations of intensity-based measurement as follows:

Flow cytometry primarily measures the fluorescence intensity of fluorophores to quantify cell surface antigens or proteins for cell type identification^{4,5} and to assess changes in physiological parameters such as pH and calcium ion concentration⁸.

However, its accuracy can be compromised by fluctuations in fluorescence intensity due to factors such as light scattering and absorption within complex cellular structures, concentration dependence and overlapping emission spectra of fluorescent molecules, and inconsistent excitation light intensity⁹.

Regarding the scatter plot in Fig. 3b: we used a single color for plotting because we measured three types of beads after mixing them. Therefore, labeling different types of beads with distinct colors in the current scatter plot was not feasible, given the uncertainty regarding which bead types are actually assigned to the subpopulations shown in Fig. 3b. To ascertain the origins of the outliers in the scatter plot, we have conducted an additional experiment, measuring each type of beads separately, and generated a scatter plot for comparison as shown below, which was also added to Supplementary Information (Supplementary Fig. 4). Consequently, we have confirmed that the outliers arose from variations in fluorescence intensity, such as doublet beads and inhomogeneous fluorescent staining, consistent with our finding detailed in the Results section on Page 5.

RFig. 1 | Distributions of different types of beads separately measured with FLIM flow cytometry (blue: 1.72 ns beads, $n = 5905$; green: 2.71 ns beads, $n = 5930$; orange: 5.54 ns bead, $n = 5984$). The same polymer beads as those used in Fig. 3 were measured. Scale bars: 20 μm . The right images show the beads randomly picked from the corresponding gray boxes in the scatter plot.

Reviewer #2's comment #3:

2. Demultiplexing (Line 102)

The author may consider including more details about this process in Supplementary Materials. In particular, how would the authors select the time window? Does the digital low-pass filtering induce any artifacts (e.g., ringing) and a reduction in spatial and/or temporal resolution?

Authors' response:

We thank the Reviewer for the comment and the suggestion. In response to his/her suggestion, we have elaborated on the demultiplexing process in the Methods section on Page 10 and Supplementary Fig. 2. In demultiplexing temporal signals, we utilized a frequency window equivalent to the modulation frequency spacing between neighboring intensity-modulated beam spots (f_s) when extracting each sub-band spectrum, which corresponds to setting a time window (i.e., temporal resolution) as the reciprocal of the frequency spacing ($1/f_s$). Additionally, we implemented digital low-pass filtering by shifting the extracted sub-band spectrum to the zero frequency position and eliminating approximately 10% of its high-frequency components. This strategy can effectively reduce crosstalk noises induced by undesired interference pairs of beam spots rather than induce artifacts such as ringing in real space (Mikami et al., *Optica* **5**, 117-126, 2018). It is important to note that this low-pass filtering can diminish temporal resolution ($1/f_s$) while not affecting spatial resolution in a typical setting of our system. We have accordingly revised the Methods section (Page 10) as follows:

Signal processing to reconstruct images from temporal signals from APDs is a key procedure for FLIM flow cytometry, which is outlined in Supplementary Fig. 2. **First, we performed Fourier transformation on the temporal signal to obtain its frequency spectrum. Then, we extracted a sub-band spectrum centered on each modulation frequency using the frequency window equivalent to the modulation frequency spacing between neighboring intensity-modulated beam spots. The extracted sub-band spectrum was low-pass filtered by shifting its sub-band to the zero-frequency position and eliminating approximately 10% of its high-frequency components, which can effectively reduce crosstalk noises induced by undesired interference pairs of beam spots while retaining spatial resolution in a typical setting of our system²⁶. After that, we performed inverse-Fourier transformation on the spectrum to obtain a complex line profile of a flowing object.**

Reviewer #2's comment #4:

3. Supplementary Fig. 5

The authors show that the lifetime increases from the center to the periphery, which is quite interesting. However, it is not clear whether this increase is due to the biological process that the authors discussed in the manuscript or due to other factors. Although fluorescence lifetime is indeed pretty resilient to environmental factors, it may be influenced by some factors (e.g., reflective index). It would be nice if the authors could show a control experiment. For example, the authors could analyze the bead's data to show an unchanged lifetime in different regions from the center to the periphery. In this way, the interpretation of Supplementary Fig. 5 can rule out some systematic errors.

Authors' response:

We thank the Reviewer for the comment and the suggestion. We have reanalyzed the data obtained from the beads used in Fig. 2, calculating their fluorescence lifetime gradient from center to periphery in the same manner as shown in Supplementary Fig. 6 (formerly Supplementary Fig. 5). As depicted in the figure below, we have found no clear fluorescence lifetime gradient within the beads unlike fluorescence intensity gradient, which would help to exclude the possibility of non-biological factors contributing to the fluorescence lifetime gradient. These results have been added to Supplementary Fig. 6. Separately, we consider that changes in the refractive index induced by DNA condensation and loosening could be one of the plausible causes of fluorescence lifetime variations. We would like to ask the Reviewer to refer to our response to Reviewer #1's comment #6 for further details. Furthermore, we would also like to ask him/her to refer to Supplementary Fig. 12 and our response to Reviewer #3's comment #6, where we have validated our findings using conventional confocal FLIM.

RFig. 2 | Distributions of fluorescence lifetime and fluorescence intensity gradients calculated using the beads data from Fig. 2. Lifetime: $n = 9496$, median = -0.25% , standard deviation (SD) = 4.07% ; Intensity: $n = 9496$, median = 54.5% , SD = 3.50% . The box plots display the median values with lines inside boxes, the first and third quartiles with box edges, 1.5 times the interquartile range (IQR) with whiskers, and outliers with dots.

Reviewer #2's comment #5:

4. Supplementary Fig. 11

Could the author add an explanation in the manuscript as to why a single modulation function can introduce the artifacts shown in the left panels in Figs. 11 a and b?

Authors' response:

We thank the Reviewer for the comment. The modulation-frequency-dependent artifacts observed in Supplementary Fig. 13a (formerly Supplementary Fig. 11a) can be attributed to the non mono-exponential fluorescence decay characteristics of the beads (PolyAn 11000006). On the other hand, the artifacts in Supplementary Fig. 13b (formerly Supplementary Fig. 11b) primarily stemmed from the fluorescence lifetime of the beads, regardless of it being non mono-exponential decay. To clarify this point, we have added the following description to the caption of Supplementary Fig. 13.

The modulation-frequency-dependent artifacts observed in **a** can be attributed to the non-mono exponential fluorescence decay of the beads, whereas those in **b** primarily stemmed from the fluorescence lifetime of the beads, regardless of it being non mono-exponential decay.

Reviewer #2's comment #6:

Reviewer #2 (Remarks on code availability):

The link is not available.

Authors' response:

We thank the Reviewer for the remark. We directly provided the codes to editors via Google Drive during the initial submission. We will resubmit the codes again.

To Reviewer #3:

We are grateful to the Reviewer for taking the time to review our manuscript and give us his/her valuable comments. We have taken all the comments into consideration and have made appropriate changes to the manuscript. Our point-by-point response appears below, in which we first echo the Reviewer's comments (shown in italic) and then respond to them. All the changes to the manuscript and Supplementary Information are highlighted in red.

Reviewer #3's comment #1:

The manuscript "FLIM flow cytometry reveals cellular characteristics hidden behind fluorescence intensity fluctuations" describes new device for flow cytometry combined with FLI detecting capability. Fluorescence Lifetime Imaging (FLI) is a powerful method, bringing more quantitative dimension into conventional fluorescent imaging and microscopy experiments. It can be combined with dyes, nanosensors, fluorescent proteins and cellular autofluorescence to measure variety of parameters, including interaction of fluorescent drugs with cellular DNA, when combined with live nuclear stains, as it is demonstrated in the manuscript with Syto 16 and doxorubicin.

Overall, it is very important to bring FLI into flow cytometry, as with this manuscript. However, flow cytometry by itself is a more destructive method, leading to cell stress, compared with e.g. label-free FLIM of live cells, organoids or live animals. Therefore, I recommend revising the manuscript to address the comments presented below.

Authors' response:

We thank the Reviewer for the comment and overall recognition of our work.

Reviewer #3's comment #2:

My major concern is about the observed finding on discriminating the subpopulations of rat glioma cells - would it be possible to validate this finding with 'conventional' FLIM microscopy?

Authors' response:

We appreciate the Reviewer for expressing his/her concern. We have examined SYTO16-stained cells derived from rat glioma using time-correlated single-photon counting (TCSPC)-based confocal FLIM (TCS SP8-FALCON, Leica Microsystems). As shown in RFig. 3a, clear cell-cell differences in fluorescence lifetime were observed. Furthermore, by applying a gate of a fluorescence lifetime of 7 ns to segregate nuclei with shorter and longer lifetimes (RFig. 3b) and analyzing their morphological features, we have found consistency with our findings in Fig. 5c (RFig. 3c). Although these validations are indirect, we consider this to be the limit of what can be achieved at the conventional microscopic level for several reasons. Firstly, the slow imaging speed of conventional FLIM practically limits the number of cells that can be analyzed under consistent sample and imaging conditions. Secondly, differences in imaging methods pose challenges to analyzing data with the same image-based parameter we used (e.g., fluorescence lifetime gradient). As shown in the confocal images of RFig. 3a (and RFig. 4a), many clear hollows (i.e., dim regions) were observed within single nuclei due to the depth sectioning capability of confocal microscopy, unlike our imaging method (Supplementary Fig. 5,8a,9). Moreover, lifetime values are different between confocal FLIM and our FLIM. To better characterize our FLIM flow cytometer, we have added the insights obtained through this validation process as follows:

In Line 262 (Page 8): **In fact, we found that the fluorescence lifetimes of SYTO16 observed with our FLIM were shorter**

than TCSPC-based confocal FLIM (Supplementary Fig. 12, Supplementary Fig. 14).

In Line 270 (Page 8): While we introduced the parameters of fluorescence intensity and fluorescence lifetime gradients for large-scale image-based profiling of heterogeneous cells in this study (Supplementary Fig. 6), it is worth noting that these parameters may not be universally applicable across all imaging methods. As shown in Supplementary Fig. 12 and Supplementary Fig. 14, TCSPC-based confocal FLIM revealed dim regions within the nuclei stained with SYTO16 due to the depth sectioning capability of confocal microscopy, a phenomenon less apparent with our FLIM (Supplementary Fig. 5, Supplementary Fig. 9). Moreover, fluorescence lifetime values were different between both methods, as mentioned above. These indicate the challenges of applying the same parameters for image analysis to images obtained with different methods. Since conventional confocal microscopy is suited for detailed observation of a limited number of cells, unlike our high-throughput FLIM flow cytometry, alternative parameters may be more suitable for its image analysis.

While we discriminated cell subpopulations in male rat glioma (Fig. 5), expanding our FLIM flow cytometer and conducting further in-depth analyses are crucial to thoroughly categorize these subpopulations into specific cell types and states. Specifically, integrating additional detection channels for immunofluorescence into our FLIM flow cytometer can enhance the resolution and specificity of subpopulation identification. This can be achieved by separating glioma cells and immune cells with the help of CD markers, and further characterizing subtypes of immune cells³. Additionally, incorporating a cell sorting system²⁵ into our FLIM flow cytometer offers another promising avenue for investigating subpopulations. This technology enables us to isolate specific cell populations and, in turn, perform downstream analyses, such as gene expression analysis, for each subpopulation.

RFig. 3 | Analysis of tumor-derived rat glioma cells imaged with time-correlated single-photon counting (TCSPC)-based confocal FLIM. **a**, Representative fluorescence lifetime images of the cells derived from a male rat glioma obtained with confocal TCSPC-based FLIM. The nuclei were stained with 2.5 μM SYTO16 for 1 hour. Scale bars: 20 μm . **b**, Scatter plot of the cells in terms of fluorescence lifetime and fluorescence intensity. $n = 640$. **c**, Comparison of morphological features for the nuclei with shorter (<7 ns) and longer (>7 ns) fluorescence lifetimes. The box plots display the median values with lines inside boxes, the first and third quartiles with box edges, 1.5 times the interquartile range (IQR) with whiskers, and outliers with dots. ES represents effect size (Cohen's d). p values were calculated with Welch's t -test (two-sided). Note that differences in segmentation algorithms used in FLIM flow cytometry and those used in image analysis for confocal microscopy may result in the absolute values of nuclear sizes obtained by both methods.

Reviewer #3's comment #3:

Apart from this, I have only minor comments:

- page 4, "non-single exponential decay" - did you mean 'non mono-exponential'?

Authors' response:

We thank the Reviewer for the comment. We have corrected it accordingly.

Reviewer #3's comment #4:

- The device is based on 488 nm laser, could you please comment on the possibility of using other wavelengths and spectrally resolved FLI with your cytometry approach in general?

Authors' response:

We thank the Reviewer for the comment. It is feasible to implement a laser with a different wavelength as a new excitation source into our system, as long as we carefully redesign the optical system, including components sensitive to wavelengths, such as acousto-optic elements and a dichroic filter. Implementing spectrally resolved FLI is also possible, in general, by adding dichroic filters or gratings to spatially separate fluorescence lights based on their wavelengths, and single-pixel photodetectors to detect them. We have added the text below to the Methods section on Page 9 (Line 315).

For our system, it is possible to implement a laser with a different wavelength as a new excitation source, as long as we carefully redesign the optical system, including components sensitive to wavelengths, such as AODs and a dichroic filter. Implementing spectrally resolved fluorescence lifetime imaging is also possible by adding dichroic filters or gratings to separately detect fluorescence light based on their wavelengths.

Reviewer #3's comment #5:

- What is the general sensitivity of the device, compared to conventional confocal FLIM (microscopy)? How the used concentrations (and long incubation times for staining) of Syto 16, Dox etc. are correlated with those used in FLIM with live cells? What about potential toxicity and effects on e.g. cell nuclei, metabolic activity etc.?

Authors' response:

We thank the Reviewer for the comment. Our fluorescence lifetime imaging system has unique sensitivity compared to

conventional laser-scanning FLIM. We have added a section describing this sensitivity on Page 16 as follows:

Our FLIM offers unique sensitivity compared to conventional laser-scanning imaging techniques. This is due to the multiplexing of image pixel acquisition in our FLIM, which allows for an increase in pixel dwell time, and thus signal strength per pixel by a factor of N_{spots} , the number of beam spots, compared to conventional techniques with the same line-scan rate²⁶. However, the superposition of shot noises from all beam spots in our FLIM leads to a decrease in the signal-to-noise ratio (SNR) by a factor of $\sqrt{N_{\text{flu}}}$, where N_{flu} represents the number of beam spots used for fluorescence excitation per line scan, assuming a uniform fluorophore distribution. As a result, the SNR improvement can be increased by a factor of $\sqrt{N_{\text{spots}}}/\sqrt{N_{\text{flu}}}$.

We interpreted the second question regarding the concentrations and incubation times as an inquiry about comparing our study to others that utilize a different (e.g., commercially available) FLIM technique, given that our study also involves FLIM with live cells. The range of doxorubicin concentrations and incubation times used in this study was determined to fall within the range used in previous research (Sparks et al., *Nature Communications* **12** 1152 (2018), Chen et al., *PLoS One* **7** e44947 (2012)), as shown in the revised Method sections on Page 12 (Line 394). Additionally, we followed the manual provided by the manufacturer (Thermo Fisher Scientific) for the range of SYTO16 concentrations and incubation times.

In relation to the toxicity of doxorubicin, the aforementioned previous research indicates that the introduction of doxorubicin results in double-strand breaks in DNA and chromatin condensation within two hours. Additionally, given that SYTO16 binds to nucleic acids, it is expected to affect cell nuclei and their conditions. However, there is currently no data available on the toxicity of SYTO16 provided by the manufacturer.

Reviewer #3's comment #6:

- Please explain 'fluorescence lifetime gradient' (page 6, referring to Supplementary Fig. 5). What do you mean by this? Can you validate it using FLIM?

Authors' response:

We thank the Reviewer for the comment. We introduced the parameter to evaluate intranuclear fluorescence lifetime distributions of many cells (e.g., 10,000 cells) imaged with our high-throughput FLIM flow cytometer. It indicates the degree of the fluorescence lifetime difference from the outer area of a nucleus to the inner area compared to the average fluorescence lifetime value of the entire nucleus. Specifically, we calculated the parameter by dividing a single nucleus into 4 ring-shaped regions and determining the value changes from the outermost contour (i.e., peripheral area) to the innermost contour (i.e., central area), as described in the Methods section on Page 13 and Supplementary Fig. 6 (formerly Supplementary Fig. 5). We found that the parameter in the nuclei of Jurkat cells stained with SYTO16 overall showed a plus value in Fig. 6. Namely, the inner areas of the nuclei exhibited longer fluorescence lifetimes than the corresponding outer areas. We considered that this is attributed mainly to the self-quenching of SYTO16 derived from the differences in DNA states (e.g., density, loosening). To validate these findings, we have obtained a fluorescence lifetime image of Jurkat cells stained with SYTO16 using the same confocal FLIM system as that used in our response to comment #2. Consequently, confocal FLIM revealed not only internuclear but also intranuclear differences in fluorescence lifetime (RFig. 4). In particular, the dim regions near the center of the nuclei showed longer fluorescence lifetimes, as evidenced by the negative correlations between fluorescence intensity and fluorescence lifetime pixel values (RFig. 4c), which is consistent with our previous findings. We would like to ask the Reviewer to refer to our responses to comment #2 and

Reviewer #2's comment 4. To clarify these points, we have added the figure below to Supplementary information and accordingly revised the Discussion section as shown in our response to comment #2.

RFig. 4 | Analysis of Jurkat cells imaged with confocal TCSPC-based FLIM. The nuclei were stained with $4 \mu\text{M}$ SYTO16 for 1 hour. **a**, Representative fluorescence lifetime image of Jurkat cells. Scale bar: $10 \mu\text{m}$. **b**, Gray-scale fluorescence intensity and fluorescence lifetime images corresponding to panel **a**. **c**, Scatter plots showing pixel distributions in fluorescence intensity and fluorescence lifetime corresponding to the segmented nuclei indicated by the arrows in panel **b**. The black dashed lines indicate linear fits based on the least square method.

Reviewer #3's comment #7:

- In general, calling the method as FLIM flow cytometry is a bit confusing. A lot of researchers mean microscopy by FLIM, and simply imaging by FLI. Some 'widefield-like' images are generated but validating with 'real FLIM' would make the manuscript much more interesting and appealing to broader audience. Please also add some discussion on pros/cons of presented method compared with FLIM and which assays can be useful on this new cytometry method.

Authors' response:

We thank the Reviewer for the comments and suggestions. We acknowledge the potential confusion, as pointed out by the Reviewer, inherent in the term “FLIM flow cytometry”. However, given the primary aim of our study, which is to integrate FLIM (i.e., microscopy) into high-throughput flow cytometry by significantly enhancing the imaging speed of FLIM, it is logically sound to designate our developed method as FLIM flow cytometry. In fact, the single-cell images obtained with our method are fluorescence lifetime microscopic images acquired in a flow cytometric configuration. Moreover, we recognize the ease of the pronunciation of “FLIM flow cytometry”. Therefore, we would like to maintain the term “FLIM flow cytometry” throughout the manuscript except for the title, which we have revised to “High-throughput fluorescence lifetime imaging flow cytometry”.

Regarding the validation with another FLIM, we would like to ask him/her to refer to our responses to comment #2 and comment #6. We believe that the validation and comparison with commercially available confocal FLIM, as added to the manuscript, have broadened its appeal and increased its robustness. We appreciate his/her suggestions again.

As for the advantages and disadvantages of our method compared to other FLIM techniques, we would like to ask him/her to refer to the Discussion section. The main advantage of our FLIM is its world’s fast imaging speed. The demonstrated image acquisition rate exceeded 10,000 images per second even though the triggering for image acquisition in flow cytometry follows a Poisson distribution (Fig. 2). This enables comprehensive fluorescence-lifetime-image-based profiling of heterogeneous cell populations, which conventional FLIM techniques fall short. On the other hand, our current system has a limitation in that it is designed to measure fluorescence lifetimes within the range of a few nanoseconds, making it challenging to accurately measure long lifetimes exceeding 10 ns. Additionally, our measurements tend to underestimate fluorescence lifetimes, as mentioned in our response to Reviewer #1’s comment #3. These points are discussed in the current version of the Discussion section.

Assays potentially useful in our method include the evaluation of drug efficacy by observing changes in the nuclear structures of cancer cells, as shown in Fig. 6. We have slightly modified the Discussion section on Page 7 (Line 233) as follows:

These results showcase that FLIM flow cytometry holds significant promise for gaining valuable insights across diverse biomedical applications **including the evaluation of drug efficacy.**

Reviewers' Comments:

Reviewer #1:

Remarks to the Author:

Reviewer's comments on the revised manuscript „High-throughput fluorescence lifetime imaging flow cytometry (#NCOMMS-24-01072A)” by H. Kanno et al.

The authors answered all my questions and carried out all the required alterations on the manuscript, which I accept. From my part, the manuscript might be published.

Reviewer #2:

Remarks to the Author:

The authors have thoroughly addressed my comments. I support the publication of the revised manuscript as is.

Reviewer #3:

Remarks to the Author:

The authors have fully addressed the criticism raised by me and by other reviewers. I find their replies satisfactory and feel that the manuscript has been improved even further. I therefore fully recommend accepting it.

Reviewer #1's comment #1:

The authors answered all my questions and carried out all the required alterations on the manuscript, which I accept. From my part, the manuscript might be published.

Authors' response:

We appreciate the Reviewer's thoughtful consideration of our revisions.

Reviewer #2's comment #1:

The authors have thoroughly addressed my comments. I support the publication of the revised manuscript as is.

Authors' response:

We thank the Reviewer for noting our revisions and providing the positive comment.

Reviewer #3's comment #1:

The authors have fully addressed the criticism raised by me and by other reviewers. I find their replies satisfactory and feel that the manuscript has been improved even further. I therefore fully recommend accepting it.

Authors' response:

We thank the Reviewer for recognizing our responses and updates to the manuscript and for the positive comment.